# Exploring dopant effects in stannic oxide nanoparticles for $CO_2$ electro-reduction to formate

Young-Jin Ko [1,9 ✉], Jun-Yong Kim[2,3,9], Woong Hee Lee [1,9], Min Gyu Kim [4], Tae-Yeon Seong[3], Jongkil Park[5], YeonJoo Jeong [5], Byoung Koun Min [1,6], Wook-Seong Lee[2], Dong Ki Lee [1,6,7 ✉] & Hyung-Suk Oh [1,7,8 ✉]

The electrosynthesis of formate from $CO_2$ can mitigate environmental issues while providing an economically valuable product. Although stannic oxide is a good catalytic material for formate production, a metallic phase is formed under high reduction overpotentials, reducing its activity. Here, using a fluorine-doped tin oxide catalyst, a high Faradaic efficiency for formate (95% at 100 mA cm$^{-2}$) and a maximum partial current density of 330 mA cm$^{-2}$ (at 400 mA cm$^{-2}$) is achieved for the electroreduction of $CO_2$. Furthermore, the formate selectivity (≈90%) is nearly constant over 7 days of operation at a current density of 100 mA cm$^{-2}$. *In-situ/operando* spectroscopies reveal that the fluorine dopant plays a critical role in maintaining the high oxidation state of Sn, leading to enhanced durability at high current densities. First-principle calculation also suggests that the fluorine-doped tin oxide surface could provide a thermodynamically stable environment to form HCOO* intermediate than tin oxide surface. These findings suggest a simple and efficient approach for designing active and durable electrocatalysts for the electrosynthesis of formate from $CO_2$.

[1] Clean Energy Research Center, Korea Institute of Science and Technology (KIST), Hwarang-ro 14-gil 5, Seongbuk-gu, Seoul 02792, Republic of Korea. [2] Electronic Materials Research Center, Korea Institute of Science and Technology (KIST), Hwarang-ro 14-gil 5, Seongbuk-gu, Seoul 02792, Republic of Korea. [3] Department of Materials Science and Engineering, Korea University, Seoul 02841, Republic of Korea. [4] Beamline Research Division, Pohang Accelerator Laboratory (PAL), Pohang 37673, Republic of Korea. [5] Center for Neuromorphic Engineering, Korea Institute of Science and Technology (KIST), Hwarang-ro 14-gil 5, Seongbuk-gu, Seoul 02792, Republic of Korea. [6] Graduate School of Energy and Environment (Green School), Korea University, 145 Anam-ro, Seongbuk-gu, Seoul 02841, Republic of Korea. [7] Division of Energy and Environmental Technology, KIST school, Korea University of Science and Technology, Seoul 02792, Republic of Korea. [8] KIST-SKKU Carbon-Neutral Research Center, Sungkyunkwan University, 2066 Seobu-ro, Jangan-gu, Suwon 16419, Republic of Korea. [9] These authors contributed equally: Young-Jin Ko, Jun-Yong Kim, Woong Hee Lee. ✉email: 091183@kist.re.kr; dnklee@kist.re.kr; hyung-suk.oh@kist.re.kr

The catalytic conversion of $CO_2$ to fuels or valuable chemical products provides a carbon-neutral cycle that can mitigate the rapid consumption of fossil resources and increasing $CO_2$ emissions[1,2]. Recently, in accordance with global $CO_2$ reduction regulations, carbon capture utilization and storage (CCUS) technology for large-scale greenhouse gas reduction and conversion to high-value-added products has been intensively researched. In particular, the electrocatalytic reduction of $CO_2$ has attracted interest owing to recent developments in electricity production from renewable energy sources such as solar and wind[3,4]. As a widely used raw material in the pharmaceutical, tanning, and textile industries that can also act as a hydrogen carrier for fuel cells[5], formate is a very attractive product of the $CO_2$ reduction reaction ($CO_2$RR). Furthermore, given the required energy input and the market price of formate, the electrochemical reduction of $CO_2$ to formate is an economically valuable process[6]. Various studies on the use of heteroatom-doped/alloy catalysts or catalyst structures with large active areas for the electrocatalytic reduction of $CO_2$ to formate have been reported[7–11]. However, a highly efficient catalyst that can meet commercial requirements for activity, selectivity, and stability has not yet been achieved.

$$CO_2 + 2H^+ + 2e^- \rightarrow HCOOH \ E^\circ = -0.12 \,(V \text{ vs RHE}) \quad (1)$$

$$2H^+ + 2e^- \rightarrow H_2 \ E^\circ = 0 \,(V \text{ vs RHE}) \quad (2)$$

Until now, metal-based catalysts have generally been used for $CO_2$ electroreduction because of their high activity and stability[12–21]. As the $CO_2$RR to formate (Eq. 1) competes with the hydrogen evolution reaction (HER; Eq. 2), inhibiting the HER is essential for obtaining a high selectivity for formate. According to Trassati's volcano plot[22], metals such as Pb[23], Bi[24], In[25], Hg[24], and Sn[23], which are generally located on the left branch of the volcano plot, exhibit high $CO_2$RR selectivity for formate. The weak metal–hydrogen bonds of these metals result in good $CO_2$RR activity for formate production.

Among these materials, Sn is the most reasonable material because the toxicity (Pb, Hg) or relatively scarcity in the earth's crust (Bi) of other typical catalyst materials limit their commercialization[26]. Considerable efforts have been focused on developing Sn-based catalysts for formate production, with recent advances including the use of gaseous $CO_2$ flow cells[27–40]. Although improvements in the catalytic activity for formate production have been achieved, the long-term durability of Sn-based catalysts under reduction conditions remains a critical issue (Supplementary Table 1). Previous research has indicated that a Sn species with a high oxidation state is key for achieving high catalytic activity for the $CO_2$RR to formate[41,42]. Bocarsly et al. observed intermediates on the Sn electrode by in situ infrared spectroscopy and suggested that the oxidized Sn surface is a catalytically active species for the $CO_2$RR[43]. However, Sn electrocatalysts with high oxidation states are reduced at high reduction overpotentials during the $CO_2$RR, resulting in the formation of a metallic phase and the loss of catalytic activity[41]. This phenomenon can be suppressed under strongly alkaline conditions, but alkaline electrolytes, such as potassium hydroxide (KOH), can be neutralized during $CO_2$RR due to the purging of $CO_2$. Therefore, it is necessary to develop alternative Sn-based electrocatalysts to ensure stability at high current densities.

Here, we studied a fluorine-doped tin oxide (FTO) nanocatalyst that not only showed high $CO_2$RR activity over a wide range of current densities but also maintained its performance for more than a week. The electrochemical $CO_2$RR performance is significantly affected by the device design and the type of purged $CO_2$[44,45]. Therefore, a homemade gaseous $CO_2$ fed flow cell was used to achieve a high current density. *In-situ/operando* analysis

was conducted under similar conditions. We found that replacing Sn–O bonds with Sn–F bonds promotes the interactions of catalyst surface and $HCOO^-$, and modified electronic structure of $CO_2$ to facilitate electron transfer. The F dopant was also revealed to play a significant role in maintaining the oxidation state of Sn at high reduction overpotentials. This work provides an advanced strategy for synthesizing cost-effective $CO_2$RR electrocatalysts with high activity and selectivity.

## Results

**Structure and physical properties of stannic oxide electrocatalysts.** $SnO_2$, fluorine-doped-$SnO_2$ (FTO), antimony-doped tin oxide (ATO), and indium-doped tin oxide (ITO) nanoparticles supported on carbon black were synthesized using a sol-gel method with hydrothermal treatment. The overall synthesis scheme is illustrated in Fig. 1a and includes the following steps:[46] (1) formation of a metal-surfactant complex, (2) hydrolysis and condensation, (3) formation of a micelle-like surfactant template with a $SnO_2$ phase, and (4) hydrothermal treatment for crystallization. The mechanism is described in more detail in Supplementary Note 1. Figure 1b, c and Supplementary Fig. 1 shows high-resolution transmission electron microscopy (HR-TEM) images of the synthesized $SnO_2$ and doped-$SnO_2$ catalysts with corresponding particle size distributions (insets, Fig. 1b, c). The $SnO_2$ and doped-$SnO_2$ samples consist of very small oxide clusters (<5 nm) with uniform distributions. All the catalysts had similar average particle sizes [2.567 nm ($SnO_2$), 2.121 nm (FTO), 2.391 nm (ITO), and 2.154 nm (ATO)], and the dopants were uniformly distributed in the doped-$SnO_2$ particles (Supplementary Figs. 2–4). On the contrary, while the $SnO_2$ catalyst without tetradecylamine (TDA) surfactant has a similar particle size compared to $SnO_2$ with TDA surfactant (2.603 nm, Supplementary Fig. 5), it aggregated to show a disordered mesoporous structure.

The $SnO_2$ and doped-$SnO_2$ nanoparticles all exhibited rutile tetragonal crystal structures, as identified by analyzing the zone axis of the images[47,48]. Furthermore, the crystal structures and average particle sizes of the synthesized nanoparticles were analyzed using X-ray diffraction (XRD) (Fig. 1d). In the XRD patterns, the (110), (101), and (211) reflections of tetragonal $SnO_2$ were observed at 2θ values of 26.3°, 33.6°, and 51.9°, respectively[46,49]. The quality of the synthesized $SnO_2$ and doped-$SnO_2$ catalysts was analyzed by thermogravimetric analysis (TGA) (Supplementary Fig. 6). All the catalysts showed a weight loss of ~20% from 30 to 500 °C owing to the removal of adsorbed water molecules and stable oxygen functional groups in the carbon support[50,51]. Under flowing $O_2$, the catalyst weight slightly increased and then decreased rapidly, which was attributed to complete oxidation of the carbon support to $CO_2$ gas after the formation of oxygen functional groups. The $SnO_2$ and doped-$SnO_2$ catalysts were both found to have oxide contents of approximately 40 wt%. The similarities in the morphologies, crystal structures, and oxide contents of the $SnO_2$ and doped-$SnO_2$ nanoparticles allowed comparisons of their catalytic activities and efficiencies for $CO_2$ reduction to formate under the same conditions.

**$CO_2$-to-formate conversion performance.** The electrochemical $CO_2$RR activities of the $SnO_2$-based catalysts were evaluated in a homemade flow cell using gaseous $CO_2$ to accelerate the $CO_2$RR while minimizing the mass transfer resistance. A detailed schematic of the flow cell is shown in Fig. 2a and Supplementary Fig. 7. The $SnO_2$-based catalysts were loaded onto a gas diffusion layer (GDL) and gaseous $CO_2$ was supplied to the cathode. An electrolyte of 1 M KOH or 1 M $KHCO_3$ was used in both the

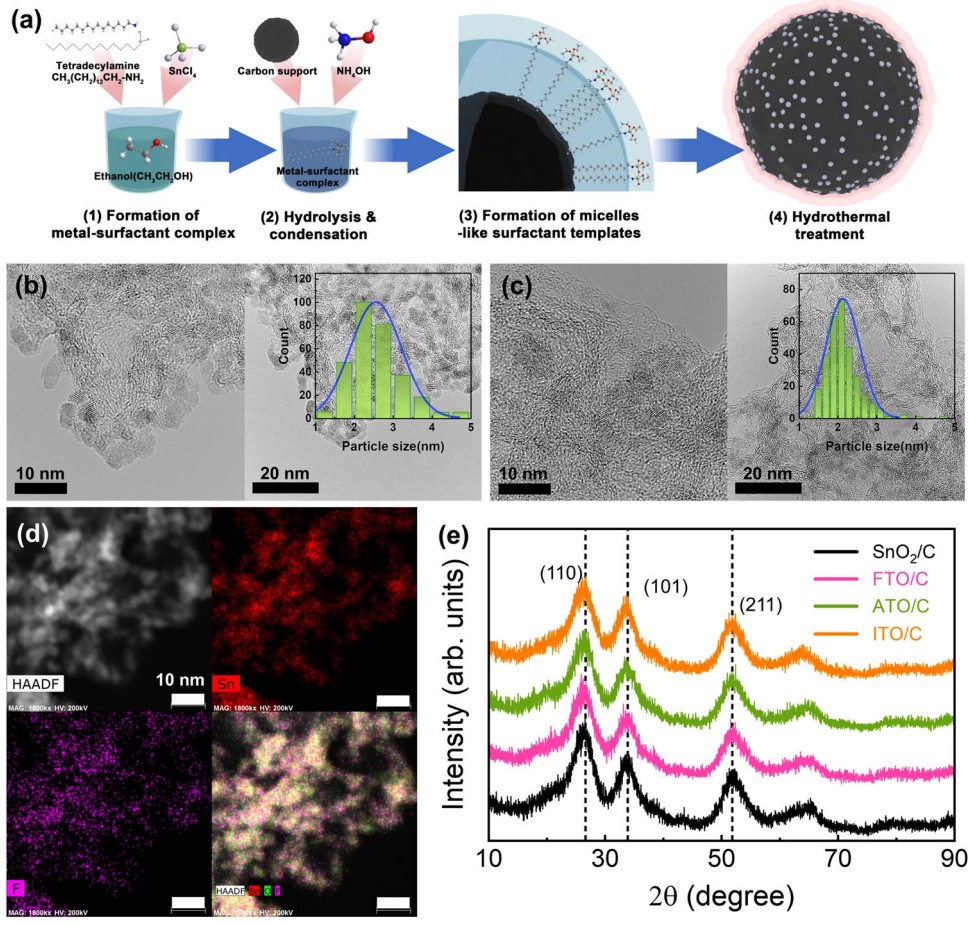

**Fig. 1 Physical properties and synthetic scheme of Sn-based catalysts. a** Synthetic pathway of well-dispersed Sn-based catalysts supported on carbon: (1) formation of metal-surfactant complex, (2) hydrolysis and condensation, (3) formation of micelle-like surfactant templates, and (4) hydrothermal treatment for recrystallization. HR-TEM images of **b** $SnO_2$/C and **c** FTO/C catalysts (Inset: particle size distributions; average particle sizes and standard deviations fitted with a Gaussian function). **d** HAADF-STEM image and its energy-dispersive X-ray spectroscopy (EDS) mapping images of Sn (red), F (magenta), and layered image combining all maps for FTO/C. The signal collecting time was 5 min. **e** Powder XRD spectra of Sn-based catalysts with various dopants. It indicated that the no phase change occurred by dopants.

cathode and anode flow channels, which were physically separated using an anion exchange membrane (AEM). Supplementary Figure 8a shows the linear sweep voltammetry (LSV) curves for $SnO_2$/C and $SnO_2$/C without tetradecylamine (TDA) using 1 M KOH as the electrolyte. $SnO_2$/C exhibited significantly lower overpotentials than $SnO_2$/C without TDA. At current densities below 300 mA cm$^{-2}$, $SnO_2$ showed a faradaic efficiency for formate (FE$_{formate}$) of more than 70% (Supplementary Fig. 8b). In contrast, for $SnO_2$/C without TDA, FE$_{formate}$ was reduced to 28.6% at 200 mA cm$^{-2}$. The maximum formate production rate of $SnO_2$/C (4.11 mmol h$^{-1}$ cm$^{-2}$) was almost three times higher than that of $SnO_2$/C without TDA (1.44 mmol h$^{-1}$ cm$^{-2}$). The effect of TDA on the $SnO_2$ particle size before and after CO$_2$RR is not substantial. However, without TDA, the $SnO_2$ particles are agglomerated (Supplementary Fig. 9: HR-TEM images of the $SnO_2$ catalyst with and without TDA). These results show that the uniformity of the $SnO_2$ particles significantly affects their CO$_2$RR activity.

To observe the effects of doping on the CO$_2$RR activity, the electrochemical CO$_2$RR activities of FTO/C, ATO/C, and ITO/C were compared with that of $SnO_2$/C (Fig. 2b, c and Supplementary Figs. 8 and 9). All the $SnO_2$-based electrodes exhibited similar LSV curves, but the electrochemical reaction products differed (Supplementary Fig. 10a). In 1 M KOH solution, FTO/C exhibited an excellent FE$_{formate}$ value of 95% at 100 mA cm$^{-2}$,

which is higher than that of ITO/C (85%), ATO/C (80%), and $SnO_2$/C (80%). Moreover, FTO/C maintained a FE$_{formate}$ value of more than 90% up to a current density of 300 mA cm$^{-2}$. The maximum partial current density and formate production rate of FTO/C were 330 mA cm$^{-2}$ and 6.31 mmol h$^{-1}$ cm$^{-2}$, respectively, which are superior to those of ITO/C (272 mA cm$^{-2}$, 5.20 mmol h$^{-1}$ cm$^{-2}$), ATO/C (242 mA cm$^{-2}$, 4.62 mmol h$^{-1}$ cm$^{-2}$), and $SnO_2$/C (215 mA cm$^{-2}$, 4.11 mmol h$^{-1}$ cm$^{-2}$). These results indicate that F doping of $SnO_2$ promotes the catalytic activity for the CO$_2$RR to formate.

To test the stabilities of FTO/C and $SnO_2$/C, 1 M KHCO$_3$ was used as the electrolyte. In 1 M KOH, the anolyte is neutralized during the electrochemical CO$_2$RR, leading to a high overpotential for OER (Supplementary Fig. 11). Considering the stability of the whole system, the CO$_2$ electrolyzer was optimized for neutral media. At 100 mA cm$^{-2}$, the FE$_{formate}$ value of FTO/C was ~90%, whereas that of $SnO_2$ was 75% (Supplementary Fig. 12). During the stability tests at a current density of 100 mA cm$^{-2}$, the FE$_{formate}$ value of $SnO_2$/C decreased significantly after several hours and the cell potential decreased slightly, showing the low stability of $SnO_2$/C. In contrast, the cell potential of FTO/C remained stable for 7 days and a FE$_{formate}$ value of ~90% was maintained (Fig. 2e, f). X-ray photoelectron spectroscopy (XPS), HAADF-STEM, and EDS after the stability tests demonstrated that the structure and chemical state of FTO/C

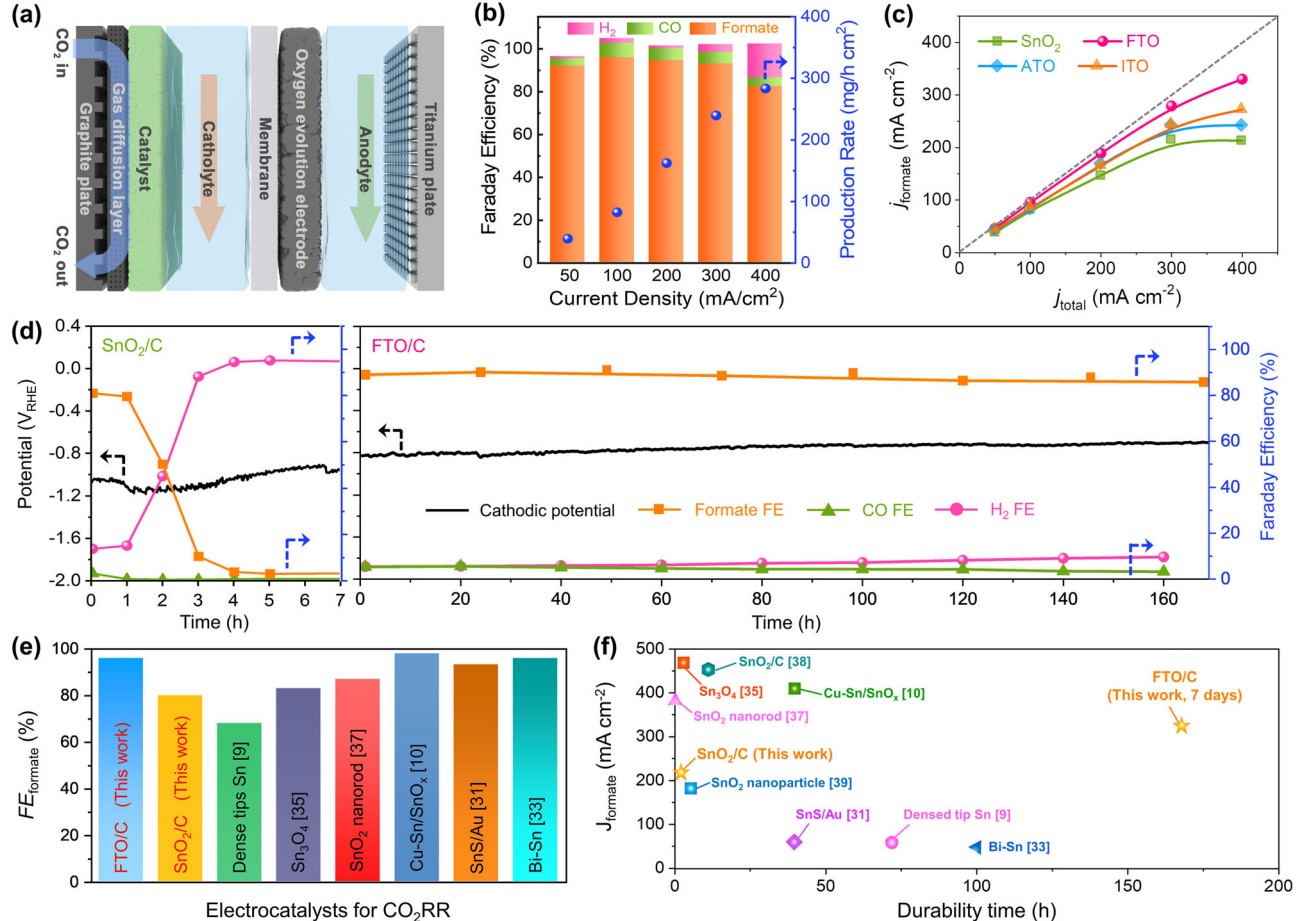

**Fig. 2 Single-cell performances of Sn-based catalysts. a** Schematic of the flow-type $CO_2$ electrolyzer using a gas-diffusion layer. **b** Faradaic efficiencies of the products and production rates of formate for FTO/C catalyst at each given current density in 1 M KOH solution. **c** Partial current densities of formate for $CO_2RR$ in the current density range of 50–400 mA cm$^{-2}$ over those of the synthesized catalysts. The error bar was calculated from three independent tests. **d** Durability test of SnO$_2$/C (left) and FTO/C (right) catalysts in the flow-type $CO_2$ electrolyzer in 1 M KHCO$_3$ solution. The faradaic efficiencies of CO, H$_2$ and formate reported were observed during the durability test. **e** FE$_{formate}$ of advanced Sn-based $CO_2RR$ catalysts. **f** Plot of the partial current density of formate (mA cm$^{-2}$) versus the durability for various Sn-based $CO_2RR$ electrocatalysts.

remained the same even after the exposure to long-term cathodic conditions (Supplementary Figs. 13 and 14). These results indicate that F doping of SnO$_2$ provides excellent long-term stability for formate production in a gaseous $CO_2$-fed flow cell. To evaluate the level of developed catalysts, the activity and stability of the FTO/C catalysts were compared with that of other Sn-based literature catalysts (Fig. 2e, f and Supplementary Table 1). The FTO/C catalysts exhibits comparable FE$_{formate}$ and current density to other best-reported catalysts, indicating enhanced intrinsic catalytic properties for $CO_2$ electro-reduction to formate. Remarkable durability of FTO/C compared to other literatures suggests that F doping improves stability of Sn catalysts required for real electrochemical formate production.

**Theoretical investigation of $CO_2$-formate conversion on the SnO$_2$ and FTO surface.** The enhanced FE$_{formate}$ of FTO was investigated through density functional theory (DFT) calculations. As the XRD patterns of the as-prepared SnO$_2$ and FTO nanoparticles did not reveal any preferred orientations, the SnO$_2$ and FTO surfaces were modeled using a four-layer slab composed of the tetragonal (110) plane. To build the FTO (110) supercell, 15% of the oxygen atoms in the SnO$_2$ (110) supercell were randomly replaced with fluorine atoms. The elementary steps of the electrochemical conversion reaction of $CO_2$ to HCOOH involving two electron pathways were described in three steps (Method).

$CO_2$ adsorption on the catalyst surface was performed to consider the onset potential difference between water and $CO_2$ reduction. The adsorbed $CO_2$ ($CO_2$*) is then converted to the HCOO* intermediate and HCOOH$_{(g)}$ in sequence with two proton-coupled electron transfers. A recent study demonstrated that, in $CO_2$ reduction current densities higher than 35 mA cm$^{-2}$, the proton can be supplied to $CO_2$* on the electrode surface by the dissociation of water molecules. At these current density regions, a huge amount of unused hydroxide ions is rapidly generated as a by-product, which results to the increase of the local interfacial pH to values above 12. This is regardless of the type of the buffering agent used[52]. In this regard, the proton for the $CO_2$ reduction reaction is assumed to be predominantly supplied by the local electrolyte.

For both the SnO$_2$ and FTO (110) surfaces, the $CO_2$ molecule was gently adsorbed on the Sn atoms. Then, the HCOO* intermediate was formed as the oxygen atoms of $CO_2$ were tightly bound to the Sn atoms. The difference between the $CO_2$ adsorption energies on the SnO$_2$ and FTO (110) surfaces was not substantial. However, approximately, a 1-eV difference was observed on the free energies for the HCOO* intermediate and HCOOH$_{(g)}$ formation steps. The FTO surface could provide thermodynamically favorable conditions for HCOO* formation compared to the SnO$_2$ surface, whereas the conditions on the SnO$_2$ surface favors the HCOOH$_{(g)}$ formation (Fig. 3 and

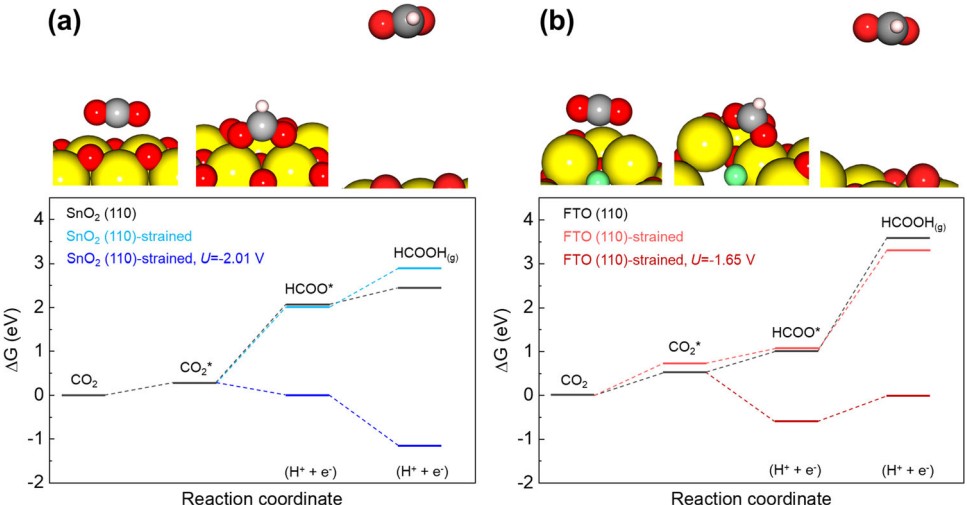

**Fig. 3 DFT calculation results of Sn-based catalysts for electrochemical $CO_2$ conversion to formate.** Free energy diagram of $CO_2$ to HCOOH conversion reaction on the **a** $SnO_2$ (110) and **b** FTO (110) surface (yellow: Sn, red: O, white: H, green: F). The strained supercells were used to simulate the structural change on $SnO_2$ and FTO nanoparticles at an applied potential of 1.0 V under $CO_2$ reduction reaction.

Supplementary Table 2). From the computational hydrogen electrode model, a minimum potential of −2.05 V was required for $SnO_2$ to overcome the activation barrier of HCOO* formation, whereas −1.79 V was needed for FTO to complete $HCOOH_{(g)}$ formation. Identical free energy calculations using strained $SnO_2$ and FTO supercells were performed to simulate the structural changes on the $SnO_2$ and FTO electrodes during $CO_2$ reduction. The in-situ EXAFS data demonstrated that the bonding distance between Sn and Sn (or O) was slightly decreased by applying a potential of −1.0 V. Consequently, the lattice constants of the $SnO_2$ and FTO supercells decreased uniformly by 1.37% and 0.33%, respectively. The observed changes in the lattice constants correspond to the average change of each bonding distance at −1.0 V (Supplementary Fig. 15 and Supplementary Table 3). Although the strained $SnO_2$ (110) surface registered a lower energy for the formation of $HCOOH_{(g)}$ by 0.42 eV, the energies for $CO_2$ adsorption and HCOO* formation were almost identical to those recorded for the $SnO_2$ surface. As such, the free energy for the HCOO* formation, which is the potential-limiting step, was not affected by the compression of the $SnO_2$ (110) crystal. In consequence, a potential of more than 2 V was still required for the strained $SnO_2$. In contrast, the strained FTO (110) surface showed a 0.27-eV lower free energy for the $HCOOH_{(g)}$ formation, while the energies for $CO_2$ adsorption and HCOO* formation slightly increased compared to that on the FTO surface. Since the $HCOOH_{(g)}$ formation is the potential-limiting step for the FTO electrode, the minimum potential to complete the $CO_2$-to-HCOOH conversion reaction decreased from −1.79 V to −1.65 V as the FTO crystal was compressed. The DFT studies suggest that doping fluorine to $SnO_2$ could alter the chemical environment of the oxide surface, making it thermodynamically stable for HCOO* intermediate formation, which is an important step in the $CO_2$-to-HCOOH conversion reaction. In addition, the compression of the crystal through the application of an external bias rendered an effective contribution only for FTO. Therefore, these factors are assumed to facilitate formate production in the FTO electrode.

**Origin of excellent durability with high current density under highly reduction potential.** In addition to the high $FE_{formate}$ of FTO/C, the high partial current density for formate at high cathodic overpotentials with excellent stability is another major

advantage of this catalyst. To reveal the origin of this behavior, *in-situ/operando* X-ray absorption near-edge structure (XANES) spectroscopy at the Sn k-edge and Raman spectroscopy were performed for $SnO_2$-based electrodes under gaseous $CO_2$RR conditions in a customized electrochemical cell with a GDL (Supplementary Figs. 16 and 17). XANES is a bulk-sensitive technique that can reveal the oxidation state of materials, whereas Raman spectroscopy is a surface-sensitive technique that can identify the chemical structure of materials. Therefore, the combination of these *in-situ/operando* spectroscopies can be used to reveal the state of materials during the $CO_2$RR. The ex-situ Sn k-edge XANES spectrum showed that both catalysts were predominantly in the quadrivalent (+4) oxidation state (Figs. 4a, b). The *in-situ/operando* XANES spectrum of $SnO_2$ exhibited a strong electrolyte-induced energy shift and fitting revealed a large fraction of metallic Sn. In contrast, the spectrum of FTO showed only a small energy shift at the reduction potential, indicating that the change in the oxidation state is extremely small. The linear combination fitting (LCF) results visually represent this trend more clearly (Fig. 4c, d).

The *in-situ/operando* Raman spectra, which show the chemical states of the surface, are shown in Fig. 4e, f. Carbon supports for the catalysts were not used for the *in-situ/operando* experiments to improve the peak intensity. The Raman peak at 630 cm$^{-1}$ [53], which is associated with the symmetric stretching of Sn–O bonds ($A_1g$ modes), was identical for $SnO_2$ and FTO, confirming the presence of oxide phase (Fig. 4a–d). This peak was observed for both $SnO_2$ and FTO at applied potentials above −0.8 V. However, for $SnO_2$, the Raman peak disappeared at a potential of −1 V, whereas for FTO, the Raman peak was still present at a potential of −1.2 V. These findings demonstrate that the surface of $SnO_2$ is converted to metallic Sn but the oxidized state of the FTO surface is maintained under high cathodic overpotentials, which is consistent with the *in-situ/operando* XANES results. Despite their similar radii, F ions have a higher electronegativity than O ions. This property would make F–Sn bonds stronger than O–Sn bonds, leading to enhanced stability of FTO. The behavior of $SnO_2$ and FTO catalysts for $CO_2$RR are summarized in Fig. 4e. $SnO_2$ catalysts exhibits good performance for formate production at low overpotential but are reduced to metallic Sn at high cathodic potentials, accelerating HER and lowering $CO_2$RR. On the other hand, FTO possess enhanced catalytic activity for $CO_2$ electro-reduction to formate by improving interaction with

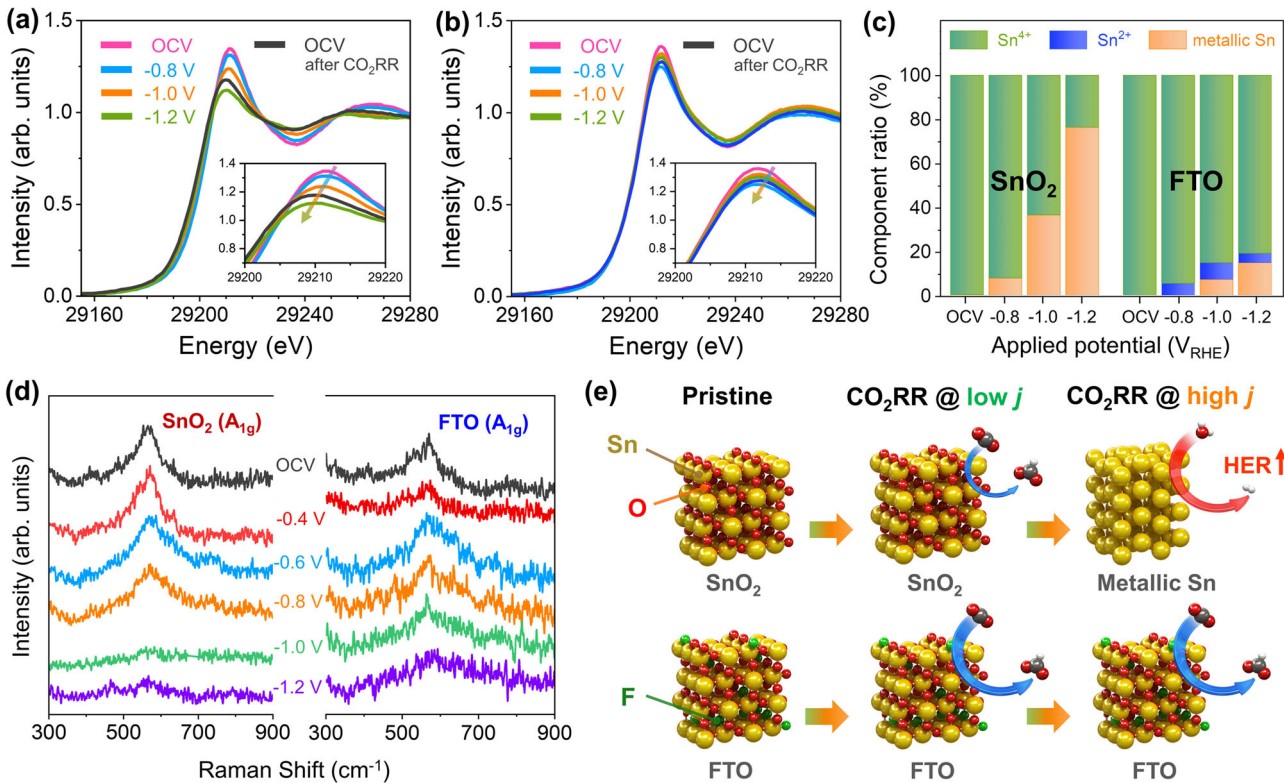

**Fig. 4 In-situ/operando spectroscopy analyses for raveling origin of durability. a, b** In-situ/operando Sn k-edge XANES spectra for **a** SnO₂/C and **b** FTO/C catalysts during CO₂RR in the flow-type electrolyzer and **c** its oxidation state distribution deconvoluted by linear combination fitting (orange: Sn, blue: Sn²⁺, and green: Sn⁴⁺). **d** In-situ/operando SER spectra obtained at constant potentials for SnO₂ and FTO catalysts without carbon supporter. Analyzed SER spectra present in the wavenumber region of 300–900 cm⁻¹. **e** Schematic illustration of reaction affinity for SnO₂ and FTO under low/high cathodic overpotential.

HCOO⁻. Furthermore, oxidation state of FTO stabilized by strong F–Sn bonds under high cathodic overpotentials contributes the enhanced current density and durability. Thus, we expect that the enhanced current density and durability of FTO could allow the development of economically feasible CO₂RR technology.

## Discussion

In summary, we obtained insights into the performance of doped-SnO₂ catalysts for the CO₂RR to formate using a combination of flow-type single-cell experiments, in-situ/operando spectroscopy, and DFT calculations. Compared with traditional SnO₂ catalysts, the nanoparticles synthesized using TDA were much smaller because this surfactant prevents particle agglomeration during SnO₂ growth via micelle formation. The high dispersion of our SnO₂ catalysts allowed for a large number of oxide species to act as electrochemically active centers. The FTO/C catalyst exhibits higher performance than other doped catalysts, achieving partial current densities for formates of up to 330 mA cm⁻² and high faradaic efficiency (95% at 100 mA cm⁻²). Notably, using the FTO catalysts, we achieve superior formate selectivity (≈90%) over 7 days of operation at 100 mA cm⁻². Based on DFT calculation, fluorine doping not only enhance the interactions between HCOO⁻ and FTO surface but also alters electronic structure of CO₂ to facilitate electron transfer. Furthermore, in-situ/operando spectroscopy suggests that in FTO catalysts, oxidation state of Sn, which significantly affects the CO₂RR activity, did not change significantly under an applied potential. These findings suggest that fluorine dopant played an important role in increasing the selectivity for formate on the FTO catalyst by modulating electronic structure of Sn and enhancing the durability by preventing

reduction under reduction potentials. Our study provides insight for designing highly active and durable electrocatalysts for the electrochemical conversion of CO₂RR to formate.

## Methods

**Preparation of SnO₂ and doped-SnO₂ catalysts.** To synthesize the SnO₂ and doped-SnO₂ catalysts, TDA (Sigma-Aldrich) was dissolved ultrasonically in a mixture of deionized water and ethanol. Then, SnCl₄ (Sigma-Aldrich) was added, and the mixture was stirred for 1 h. This suspension was ultrasonically blended with carbon black powder, and then ammonium hydroxide solution (NH₄OH, Sigma-Aldrich) was added dropwise, followed by stirring for 30 min. Subsequently, the suspension was refluxed at 80 °C for 72 h. The reaction mixture was cooled to room temperature, filtered, and washed several times with an ethanol solution. To remove excess TDA, the as-prepared SnO₂ catalyst was transferred to a glass-lined stainless-steel autoclave and hydrothermally treated at 120 °C for 24 h. A detailed description of the formation of the tin ethbutoxide intermediate in the ethanol solution and the formation of SnO₂ through the hydrolysis reaction is described in Supplementary Note 1. ATO, FTO, and ITO were synthesized using the same procedure, except that the composition of the metal precursor was varied by adding antimony acetate (C₆H₉O₆Sb, Sigma-Aldrich), ammonium fluoride (NH₄F, Sigma-Aldrich), and indium chloride (InCl₃, Sigma-Aldrich), respectively. Varying amounts of the dopant salts were added to 712 mg of SnCl₄ to synthesize the doped-SnO₂ sample. In ATO, 25.0 mg C₆H₉O₆Sb was used, in FTO, 19.48 g NH₄F, and in ITO, 109.6 g InCl₃. Antimony and indium were doped in the form of mixing Sb₂O₃ or In₂O₃ with SnO₂ during hydrolysis and condensation, respectively. On the other hand, fluorine was incorporated through oxygen substitution[54].

**Preparation of SnO₂-based catalyst electrodes for the CO₂RR.** A catalyst ink was prepared by ultrasonically mixing 5 wt% of ionomer solution (Dioxide, 15 wt% target of catalyst) and SnO₂-based catalyst powder (30 mg) with ethanol (2 mL). The SnO₂-based catalyst electrodes were fabricated by spraying the prepared catalyst ink was onto a GDL (Sigracet 39 BB, SGL Carbon) at 70 °C. The electrode area was 2 cm² and the loading of SnO₂ was fixed at 0.5 mg cm⁻².

**Preparation of Fe-Ni foam electrodes for the oxygen evolution reaction (OER in alkaline).** Fe-Ni foam electrodes for the OER were fabricated by a simple dip-

coating method. The Ni foam was washed with deionized water and dried under nitrogen. After dipping in a 0.125 M FeCl₃ solution, the Ni foam was removed and dried in a convection oven at 70 °C. The FeCl₃-coated Ni foam was then activated using a three-electrode system in 1 M KOH at a current density of 100 mA cm⁻² for 10 min. A graphite rod and Hg/HgO electrode were used as the counter and reference electrodes, respectively.

**Preparation of IrO₂/Pt coated Ti-foam electrodes for the oxygen evolution reaction (OER in neutral)**. A catalyst ink was prepared by ultrasonically mixing 5 wt% of ionomer solution (Nafion, 10 wt% target of catalyst) and commercial iridium oxide catalyst powder (Alfa Aesar, 30 mg) with ethanol (2 mL). The electrodes were fabricated by spraying the prepared catalyst ink was onto a Pt coated Ti foam at 70 °C. The electrode area was 2 cm² and the loading of IrO₂ was fixed at 1 mg cm⁻². The reason of IrO₂ catalyst for OER in neutral electrolyte was described in Supplementary Note 2.

**Electrochemical CO₂RR flow cell tests**. A detailed schematic of the flow cell used for evaluating the electrochemical CO₂RR performance is shown in Supplementary Fig. 7. The fabricated SnO₂-based catalyst electrodes were used as the cathode. The active area of each electrode was 2 cm², and CO₂ as the reactant gas was fed into the serpentine flow field channel on the cathode side at a flow rate of 50 sccm. The electrolyte solution of 1 M KOH or 1 M KHCO₃ was supplied to both the anode and cathode sides using a pump. An AEM (Dioxide Materials, X37-50 Grade RT) was used to separate the anode and cathode flow channels. All electrochemical tests were conducted using a VSP potentiostat (BioLogic, VMP3B-10), which was suitable for measurements up to 10A. The reference electrode (Ag/AgCl, 3.5 M KCl) was inserted in the cathode flow line to measure and control the cathode potential.

The Faradaic efficiency of the catalyst was measured using GC and IC where points were taken at 18 min-intervals. The formate concentrations on the catholyte and anolyte were measured to calculate the total production of formate. The composition of the outlet gas was measured using gas chromatography (GC, Agilent 7890A). The inlet of the GC, fitted with a water trap to prevent water from entering the GC, was connected to the cathode outline. Ultrahigh-purity helium gas (99.9999%) was used as the carrier gas. A flame ionization detector and thermal conductivity detector were used to detect carbon-based gases (CO, CH₄, and C₂H₄) and hydrogen gas, respectively. A methanizer was used to enhance the detection of CO. The faradaic efficiency of each product was calculated using the following equation:

$$\text{FE}_{\text{product}}(\%) = \frac{i_{\text{product}}}{i_{\text{total}}} \times 100 = \frac{V_{\text{product}} \times Q \times \frac{2Fp}{RT}}{i_{\text{total}}} \times 100 \qquad (3)$$

Where $F$ is the Faraday constant (96485 C mol⁻¹), $Q$ is the flow rate of products, $T$ is room temperature (298 K), and $R$ is the ideal gas constant (8.314 J mol K⁻¹).

The total current was measured using a VSP potentiostat. The peaks in the GC chromatogram were used to determine the volumes of specific products, which allowed the partial current density of each product to be calculated. Ion chromatography (IC) coupled with inductively coupled plasma optical emission spectroscopy (Thermo Scientific, Dionex ICS-5000⁺ HPIC) was used to monitor formate, and FE_formate was calculated using the following equation:

$$\text{FE}_{\text{formate}}(\%) = \frac{i_{\text{foramte}}}{i_{\text{total}}} \times 100 = \frac{C_{\text{formate}} \times N \times F}{i_{\text{total}}} \times 100 \qquad (4)$$

where $N$ is the number of electrons transferred and $F$ is the Faraday constant. The peak in the IC chromatogram was used to determine the concentration of specific products.

**Physical characterization**. The microstructures of the synthesized catalysts were observed using HR-TEM (FEI Co, Titan 300 kV), and the elemental distribution in each catalyst was obtained using energy-dispersive X-ray spectrometry (FEI Co, Talos 200 kV). The nanoparticle size distributions were analyzed using ImageJ software, and the obtained profiles were fitted using a Gaussian function. Ten HR-TEM images were taken for each oxide sample, from which 300 particles were selected for particle size measurements. The surface chemistry of the catalysts was evaluated using X-ray photoelectron spectroscopy (Ulvac Co., PHI 5000 Versap-robe). Wide-angle XRD (Rigaku D-max/2500-PC, Cu-Kα radiation) was used to investigate the crystal structure and identify the nature of the oxides. TGA (TA Instruments, Q600 SDT) of the catalysts was performed from room temperature to 800 °C at a rate of 10 °C min⁻¹. This analysis was performed in a N₂ atmosphere up to 500 °C, and O₂ gas was injected thereafter.

**In-situ/operando X-ray absorption spectroscopy (XAS)**. The Sn k-edge hard-X-ray absorption spectroscopy (XAS) spectra of the synthesized catalysts were recorded at the 10C beamline of the Pohang Acceleration Laboratory (PAL). The setup for the in-situ/operando hard-XAS measurements with a homemade electrochemical single cell is shown in Supplementary Fig. 16. For operando XAS measurements, a 1-cm² hole was made in the anode and cathode bipolar plate and covered with Kapton film to allow passage of the X-rays. The operational conditions were the same as in the single-cell tests, and 1M KHCO₃ was used as the catholyte and anolyte. Before the XAS measurements, the electrode was stabilized

for 5 min at each potential. The hard-XAS analysis was carried out in the fluorescence collection mode using a Si (311) monochromator. The hard-XAS spectra were calibrated using Sn foil to ensure a zero shift in the edge energy. The XANES data were fitted using the Athena software (Demeter ver. 0.9.20). To maintain consistency in the analysis, the height of the arctangent function corresponding to the transition to the continuum level was set to one.

For the EXAFS analysis, Artemis (also implemented in Demeter ver. 0.9.20) software was utilized after the processing of data using the Athena software. The background signal was removed to extract the EXAFS signal for $R_{\text{bkg}} = 1.0–1.1$ Å. The EXAFS data were transformed using the Kaiser–Bessel function. The many-body reduction factor ($S_0^2$) for Sn was determined to be 0.86 from the EXAFS curve fit of the Sn foil. The statistical quality of the curve fit to the proposed models can be determined from the R-factor and $\chi^2$ function available in the refinement.

**In-situ/operando surface-enhanced Raman spectroscopy**. The in-situ/operando surface-enhanced Raman spectroscopy (SERS) measurements for the gas-phase CO₂RR were performed using a homemade electrochemical three-electrode cell with a GDL, as shown in Supplementary Fig. 17. The excitation light source was a Nd:YAG laser (532 nm). A platinum wire was used as the counter electrode, Ag/AgCl (3.5 M KCl) was used as the reference electrode, and the catalyst-loaded GDL was used as the working electrode. For the CO₂RR, 100 sccm CO₂ was supplied to the catalyst-loaded GDL. The homemade electrochemical cell was filled to a thickness of 5 mm with 1 M KHCO₃ as the electrolyte. The electrochemical experiments were controlled using an IVIUM CompactStat.h potentiostat/galvanostat.

**DFT calculations**. DFT calculations were performed using the Quantum ESPRESSO package[55,56]. Geometry optimization was performed using the Perdew–Burke–Ernzerhof functional[57] with the projector-augmented wave pseudopotentials[58,59]. The Grimme's D3 (DFT-D3)[60] method was used to account for the van der Waals dispersion correction. A kinetic energy cutoff of 500 eV was used with a plane-wave basis set. Gaussian smearing was applied with a smearing width of 0.1 eV. The geometries were fully relaxed until the residual force on the atoms converged to 0.01 eV/Å. The SnO₂ (110) supercell was modeled using a unit cell of the tetragonal space group (mp-856) obtained from the Materials Project[61]. The four layers of the primitive unit cell that are cleaved to the (110) plane were expanded six times ($2 \times 3 \times 4$), yielding a supercell of 24 Sn and 48 O atoms with a lattice constant of 6.83 Å × 9.73 Å × 13.67 Å. To build the FTO (110) supercell, seven O atoms (~15%) were randomly replaced with F atoms. To simulate the SnO₂ and FTO crystal conditions under an applied potential of −1.0 V, the lattice constants of the SnO₂ and FTO (110) bulk were uniformly reduced by −1.37% and −0.33%, respectively (Supplementary Fig. 15 and Supplementary Table 1). The lattice constants of strained SnO₂ and FTO (110) bulk supercell were 6.74 Å × 9.60 Å × 13.48 Å and 6.81 Å × 9.70 Å × 13.62 Å, respectively. The Brillouin zone was sampled with a Monkhorst–Pack k-point mesh of ($5 \times 4 \times 2$) for bulk SnO₂ and FTO (110) supercells. To build the slab structure, a 20-Å vacuum gap was added along the c-axis of the stabilized supercells. The bottom two layers of the slab were fixed, while the top two layers were allowed to relax during geometry optimization. The Brillouin zone was sampled with a Monkhorst–Pack k-point mesh of ($5 \times 3 \times 1$) for SnO₂ and FTO (110) slab supercells.

The elementary steps of the electrochemical conversion reaction of CO₂ to HCOOH involving 2 electron pathway are described as follows:

$$CO_2 + * \rightarrow CO_2* \qquad (5)$$

$$CO_2* + H^+ + e^- \rightarrow HCOO* \qquad (6)$$

$$HCOO* + H^+ + e^- \rightarrow HCOOH + * \qquad (7)$$

where * represents the surface sites for molecule adsorption. The change in Gibbs free energy ($\Delta G$) at 298 K and 1 atm was calculated thorough $\Delta G = \Delta E + \Delta ZPE - T\Delta S$, where $\Delta E$ is the total electronic energy obtained from the DFT optimization, $\Delta ZPE$ is the change in the zero-point energies, $T$ is the temperature, and $\Delta S$ is the change in entropy. The computational hydrogen electrode model[62] was applied to calculate the chemical potential of proton/electron pairs, which is equal to the half of the chemical potential of H₂ gas under standard conditions and electrons with an applied bias of $U$ ($-eU$). The pH contribution is considered by adding $k_B T \times \ln 10 \times pH$ to $\Delta G$, where $k_B$ is the Boltzmann constant. The $ZPE$ and $S$ of the molecules and adsorbates were determined from the calculated vibrational frequencies and NIST database[63,64], where all vibrations were treated in the harmonic oscillator approximation. The $ZPE$ and $S$ data are listed in Supplementary Tables 2 and 5, respectively.

## Data availability
Source data are provided with this paper.

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

## Acknowledgements

This work was supported by institutional program grants from the Korea Institute of Science and Technology and "Carbon to X Project" (Project No. 2020M3H7A1098229) through the National Research Foundation (NRF) funded by the Ministry of Science and ICT, Republic of Korea. This research was also supported by the National Research Council of Science & Technology (NST) grant by the Korean government (MSIT) (No. CAP21011-100), National Research Foundation of Korea (NRF) grant funded by the Korean government (MSIT) (NRF-2021R1A2C2093467), and National Supercomputing Center with supercomputing resources including technical support (KSC-2021-CRE-0284). We also acknowledge Advanced Analysis Center at KIST for the TEM and Raman measurements. We wish to thank 10C PAL beamline for measuring the hard X-ray absorption spectroscopy (XAS).

## Author contributions

H.-S.O. and Y.-J.K. conceived the idea, designed the experiments, and supervised the work. D.K.L. designed/simulated the DFT calculations. J.-Y.K. synthesized the catalysts, analyzed the data, and wrote the manuscript. W.H.L. conducted the electrochemical and in-situ experiments and wrote the manuscript. M.G.K. performed XANES analysis. J.P., Y.J., and B.K.M. contributed to the electrochemical analysis. T.-Y.S. and W.-S.L. contributed to the catalyst synthesis. All authors reviewed the manuscript and agreed with its content.

## Competing interests

The authors declare no competing interests.
