## [Peer Review File · Nature Communications]

Title: Exploring dopant effects in stannic oxide nanoparticles for CO₂ electro-reduction to formateREVIEWER COMMENTS

Reviewer #1 (Remarks to the Author):

The present paper deals with electrocatalysis of CO₂ reduction to formate. This topic is of great interest and there have been numerous papers published on the topic. Doping SnO₂ as presented here is a new concept and the results are promising on a small scale (2 cm² electrode area). Please consider the following suggestions for a revised manuscript:

Line 39: it is confusing what is the 100 mA/cm² current density and the partial current density of 330 mA/cm²? Please rewrite and clarify.

Line 79 – 80: add the pH dependent equilibrium potentials for the 2 reactions.

Line 102: what does it mean 'KOH cannot be fed indefinitely'? Which electrolyte can be fed indefinitely? It is not clear what is the meaning.

Line 138-144 and 357-360: even though the preparation method has been described in a previous reference, please give a bit more detail about the specific preparation method to better understand the effectiveness. How much NH₄F was added? Etc.

Figure 2b: indicate after how many minutes or seconds of electrolysis is the faradaic efficiency is reported?

Is there an influence of gas to liquid flow rate on performance?

Reviewer #2 (Remarks to the Author):

This research shows that the fluorine-doped tin oxide catalyst exhibited high current density and selectivity toward formate with superior stability. Through in situ investigation, the high oxidation state of Sn maintained by the doped fluorine is the key factor for the enhancement. Nonetheless, some of the key statements must be addressed.

1. Figure 2f is wrong, the authors test the stability at 100mA cm⁻² under 1 M KHCO₃ condition.

However, in figure 2f, the J_{formate} is about 300 in which may be the results extracts from the KOH condition. The authors should correct it in the revised manuscript. Moreover, the authors ascertain that the stability test was carried out under 1 M KHCO₃ condition since it is more stable under working conditions owing to a small change in pH. Indeed, 1 M KHCO₃ with higher buffer capacity leads to fewer pH differences between the local environment and the bulk solution, yet according to the previous report, the local pH changes dramatically under high current density (ref: Energy Environ. Sci., 2019, 12, 1442-1453). The dramatic change of pH in both 1 M KHCO₃ and 1 M KOH implies that 1 M KHCO₃ is not stable during CO₂RR. Moreover, the local pH will reach a steady-state under a constant current. Accordingly, the reason for adopting 1 M KHCO₃ is not convincing. I consider that the author should conduct the stability test under 1 M KOH.

2. I cast doubt on the simulation that the formate conversion process involves proton transfer from water to CO₂ since the local pH is alkaline, as revealed by previous reports (ref: Energy Environ. Sci., 2019, 12, 1442-1453). Moreover, the local pH of 1 M KHCO₃ under working conditions is higher than

that of the 1 M KOH condition, which was revealed by using in situ observation (J. Am. Chem. Soc. 2020, 142, 15438–15444). Accordingly, the assumption of proton transfer from water to CO₂ is wrong.

3. For the theoretical calculations, the results should be based on the experimental evidence obtained from the in situ observations since the catalyst undergoes structural evolution during the reaction, as revealed by the in situ XAS and Raman results. As a result, I will suggest the authors should provide a correct model based on the in situ measurements.

4. The authors should provide EXAFS results extracted from XAS. I believe that if F could stabilize Sn, the local structure may retain the initial situation.

5. The authors suggested that the valence state of Sn of the FTO catalyst is oxidized under high cathodic potentials. However, according to previous reports, the oxidized feature under cathodic potential may result from the sluggish structural transformation of Sn. It may transform into metallic Sn under long-term observation. Based on this reason, I think the authors should provide a long-term observation to ascertain the oxidized Sn is stable under cathodic conditions.

6. In my opinion, the FTO/C catalyst would transform into metallic Sn after days of operation. Based on this assumption, only a trace amount of fluorine could be detected after the stability test. To resolve my concern, structural characterization such as HAADF-STEM and EDS results of post-reaction catalysts should be provided. If fluorine of FTO/C could be detected by using EDS after the reaction, it could help the authors to strengthen their claim.

7. The preparation procedure of SnO₂ and doped-SnO₂ catalysts is ambiguous. The authors should provide the exact formula in the experimental section.

8. According to LSV and FE results, SnO₂ with TDA during the synthesis process shows higher activity and selectivity toward formate. The authors claim that the particle size and uniformity of SnO₂ have significant effects on the CO₂RR activity. However, the particle size of SnO₂ with/without TDA is similar. I consider that the authors should provide more evidence such as TEM images for post-reaction catalysts to ascertain the claim for the contradiction.

To sum up, I think that the author should prove Sn is in a high oxidation state and fluorine retain under long-term examination. Moreover, the DFT calculation should be revised since it is based on an inappropriate model. As for present manuscript, this research falls short of being published in Nat. Commun.

Reviewer #3 (Remarks to the Author):

This manuscript deals with the preparation and characterization of SnO₂ catalysts doped with fluorine, which are then assessed when used in electrodes for the electrochemical reduction of CO₂ to formate. The most relevant results are related to the improved performance of fluorine-doped SnO₂ electrocatalysts, which show excellent Faradaic efficiencies at reasonably high current density (100 mA cm⁻²) during 7 days of operation. The manuscript addresses a hot topic, and it may report interesting new results, including observations from in situ/operando spectroscopies. However, I feel there are

some aspects and mistakes that deserve clarification and improvement before it could be eventually accepted for publication:

1. Page 3, lines 75-76: "Various studies have been devoted to the electrocatalytic reduction of CO₂ to formate,⁷⁻¹¹". Why have references 7-11 been selected as "studies devoted to the electrocatalytic reduction of CO₂ to formate?" Some brief justification of their relevance should be provided to show they have not been chosen randomly.
2. Page 3, line 76: "...highly efficient catalyst...": Great emphasis is placed here on the nature of the catalyst, which has indeed a great influence. However, some recent studies also show that not just the nature of the catalysts, but also the configuration of the electrode (i.e. the form in which it is used in the cathode) and the reactor design can play an important role in the performance of electrocatalytic reduction of CO₂ (see e.g. Chem. Eng. J. 364 (2019) 89–101; Chem. Eng. J. 405 (2021) 126965). It could be interesting to strengthen the introduction briefly discussing this idea.
3. Page 3, lines 89-90: "Sn is most reasonable materials because the toxicity (Pb, Hg) or high price (Bi) of catalyst materials limits commercialization". Have the authors checked that the price of Bi particles is really higher than Sn particles? A quick check in suppliers like Merck or Sigma-Aldrich shows that the price of Bi powders per gram is similar, or even lower, than the price of Sn powder of the same particle size, and if I am not wrong, the cost of Sn per kg is several times higher than that of Bi. Authors should carefully check this, and if necessary, correct the statement.
4. Page 14, lines 363-364: In the "Preparation of SnO₂-based catalyst electrodes for the CO₂RR", it is only stated that "A catalyst ink was prepared by ultrasonically mixing 5 wt% of ionomer solution (Dioxide, 15 wt% target of catalyst)..." but the nature of the ionomer was not clear to me. Which compound was used as binder and why?
5. Page 14, lines 366-367: Also regarding the preparation of the SnO₂-based cathodes, the loading of catalyst used can play an important role in the performance of the electrode. How and why was the loading fixed at 0.5 mg cm⁻²? This is an important aspect that deserves further explanation.
6. Different electrodes were used as anodes for the OER in neutral and in alkaline medium. It could be interesting to include, in the Supporting Information, the reasons to justify this decision.
7. It may difficult for the reader to know exactly the different catalysts and electrodes that were tested in the study. I suggest to clearly identify them in "4. Experiment methods", at the beginning of subsection "Electrochemical CO₂RR flow cell tests".
8. Page 14, lines 387-388: The schematic of the flow cell is not in Figure S6, but in Figure S7. Please correct.
9. Page 14, lines 388-389: "The fabricated SnO₂-based catalyst electrodes were used

as both the cathode and anode.” What do the authors mean with “both as cathode and anode”? How were the Sn-based electrodes used as anodes?

10. The way formate was collected and analyzed was not clear to me. In page 15 it is only mentioned that “IC (...) was used to monitor formate (...)” But which liquid streams did the authors actually analyze? Was formate obtained in the anolyte stream, in the catholyte or in both? My concern is that the use of an anion exchange allows the pass of anions, which means that HCOO⁻ formed in the cathodic compartment should cross the membrane and be collected in the anolyte, but perhaps only partially, and some formate product may also be found in the catholyte. I feel this should be clearly clarified in the manuscript.

11. Page 16, line 433. “...is shown in Figure XX.” Please revise.

12. Page 17. The selection of the most appropriate functionals and potentials for the DFT calculations is not trivial. How and why were these chosen for this specific application?

13. Figure 1b and c: Considering that, according to the insets with particle size distributions, the average particle size seems to be around 2nm, are scales of 10 or 20 nm the most adequate to show?

14. Figure 2 and Figure S7. As shown in the schematic illustration, in the anodic compartment, an “Oxygen Evolution Electrode” and a “Titanium Plate” are represented. Why is the titanium plate used? If there is already an OER electrode, which, as described in page 14, already includes metallic foams that can act as current collector, why is the additional titanium plate needed?

15. Fig S8: Particularly at the highest current density studied (400 mA cm⁻²), the sum of FEs is further from approaching 100% than at lower current densities. Do the authors have any explanation for that?

16. Another important concern is related to the reproducibility of the results obtained. For example, were the experiments in the flow cell reported in Figure 2 carried out more than once? In that case, which values of deviation were observed e.g. in the results of Faraday efficiencies and production rates? Some indication of the reproducibility of the results (e.g. including error bars with standard deviation) must be provided.

17. The way the experimental results are presented and discussed on “CO₂-to-formate conversion performance” should be improved. In line with a previous comment, different types of catalysts, and even different conditions are mentioned (e.g. type of electrolyte: KOH or KHCO₃), but I feel that the global picture of which combinations were actually tested was not clearly shown. Perhaps, a good way to clearly show this, could be to include in the Supporting information a Table with all the tests carried out with their different experimental conditions and results.

18. The “formate production rate” (page 194, figure 2 and others) is expressed as “mg h⁻¹ cm⁻²” But does the mass unit represent only formate anion (HCOO⁻) or the corresponding salt? To avoid any ambiguity,

I suggest to convert all these values and report the production rates with units of “mol”.

19. Line 219, Table S1, figures 2e and 2f: Which were the criteria for considering particularly these studies using Sn-based electrodes? There are many other relevant studies in the literature using Sn-based electrodes for the electroreduction of CO₂ to formate that could have also been included, so again, the criteria used for selecting these references in Table S1 should be explicitly provided to show they have not been chosen randomly.

Authors' response to the reviewer's comments

Reviewer: 1

Recommendation: The present paper deals with electrocatalysis of CO₂ reduction to formate. This topic is of great interest and there have been numerous papers published on the topic. Doping SnO₂ as presented here is a new concept and the results are promising on a small scale (2 cm² electrode area). Please consider the following suggestions for a revised manuscript:

Comment 1: Line 39: it is confusing what is the 100 mA/cm² current density and the partial current density of 330 mA/cm²? Please rewrite and clarify.

Response: We appreciate the reviewer's valuable comments. The maximum Faraday efficiency of the FTO catalyst was 95% at 100 mA cm⁻². The maximum partial current density of the FTO catalyst was 330 mA cm⁻² at 400 mA cm⁻². These results were obtained at different current density regions. To clarify this point, the manuscript has been revised accordingly.

Main manuscript Page 2, Line 5: Here, using a fluorine-doped tin oxide (FTO) catalyst, a high Faradaic efficiency for formate (95% at 100 mA cm⁻²) and a maximum partial current density of 330 mA cm⁻² (at 400 mA cm⁻²) was achieved for the electroreduction of CO₂.

Comment 2: Line 79 – 80: add the pH dependent equilibrium potentials for the 2 reactions.

Response: Thank you for this valuable comment. Following the reviewer's advice, we have added the standard equilibrium potentials (E°) against the reversible hydrogen electrode (V vs. RHE) for the two reactions. From these potentials, E° at different pH values can be calculated.

Main manuscript Page 3, Line 18:

Comment 3: Line 102: what does it mean 'KOH cannot be fed indefinitely'? Which electrolyte can be fed indefinitely? It is not clear what is the meaning.

Response: We appreciate the reviewer's valuable comments. KOH is a strong alkaline electrolyte. However, during CO₂RR, it can be neutralized by the purged CO₂. To maintain the alkaline conditions at the anode and the cathode, KOH was continuously fed. However, no electrolyte can be indefinitely fed. We hope that this clarifies the statement. The following was the revised sentence in the main manuscript to address this point.

Main manuscript Page 4, Line 10: "..., but alkaline electrolytes, such as potassium hydroxide (KOH), can be neutralized during CO₂RR due to the purging of CO₂.

Comment 4: Line 138-144 and 357-360: even though the preparation method has been described in a previous reference, please give a bit more detail about the specific preparation method to better understand the effectiveness. How much NH₄F was added? Etc.

Response: Thank you for bringing this to our attention. Following the reviewer's advice, the synthesis method for doped-SnO₂ catalysts was discussed in detail at the Experimental Methods section.

Main manuscript Page 14, Line 16: Varying amounts of the dopant salts were added to 712 mg of SnCl₄ to synthesize the doped-SnO₂ sample. In ATO, 25.0 mg C₆H₉O₆Sb was used, in FTO, 19.48 g NH₄F, and in ITO, 109.6 g InCl₃. Antimony and indium were doped in the form of mixing Sb₂O₃ or In₂O₃ with SnO₂ during hydrolysis and condensation, respectively. On the other hand, fluorine was incorporated through oxygen substitution.

Comment 5: Figure 2b: indicate after how many minutes or seconds of electrolysis is the faradaic efficiency is reported? Is there an influence of gas to liquid flow rate on performance?

Response: We are thankful to the reviewer for this valuable comment. The Faradaic efficiency for each current point was taken at 18 min-intervals. This information has been added to the Experimental section. In the proper range, the gas and liquid flow rates have negligible effects on the catalytic performance. However, when the flow rate of the liquid is much higher than that of the gas, the former crosses and blocks the GDE. This phenomenon significantly reduces CO₂RR performance. Conversely, the gas crossovers to the liquid flow channel. Crossing over of the gas causes noise and inhibits ion transfer, which also reduces CO₂RR performance.

Main manuscript Page 15, Line 24: The Faradaic efficiency of the catalyst was measured using GC and IC where points were taken at 18 min-intervals.

Reviewer: 2

Recommendation: This research shows that the fluorine-doped tin oxide catalyst exhibited high current density and selectivity toward formate with superior stability. Through in situ investigation, the high oxidation state of Sn maintained by the doped fluorine is the key factor for the enhancement. Nonetheless, some of the key statements must be addressed.

Comment 1: Figure 2f is wrong, the authors test the stability at 100 mA cm^{-2} under 1 M KHCO_3 condition. However, in figure 2f, the J_{formate} is about 300 in which may be the results extracts from the KOH condition. The authors should correct it in the revised manuscript. Moreover, the authors ascertain that the stability test was carried out under 1 M KHCO_3 condition since it is more stable under working conditions owing to a small change in pH. Indeed, 1 M KHCO_3 with higher buffer capacity leads to fewer pH differences between the local environment and the bulk solution, yet according to the previous report, the local pH changes dramatically under high current density (ref: Energy Environ. Sci., 2019, 12, 1442-1453). The dramatic change of pH in both 1 M KHCO_3 and 1 M KOH implies that 1 M KHCO_3 is not stable during CO_2RR . Moreover, the local pH will reach a steady-state under a constant current. Accordingly, the reason for adopting 1 M KHCO_3 is not convincing. I consider that the author should conduct the stability test under 1 M KOH .

Response: Thank you for this insightful comment. In accordance to the reviewer's comments, Figure 2f was duly revised. Thank you for pointing this out. Your comment has significantly helped us improve the manuscript.

Figure 2. Single-cell performances of Sn-based catalysts. (a) Schematic of the flow-type CO₂ electrolyzer using a gas-diffusion layer. (b) Faradaic efficiencies of the products and production rates of formate for FTO/C catalyst at each given current density in 1 M KOH solution. (c) Partial current densities of formate for CO₂RR in the current density range of $50\text{--}400\text{ mA cm}^{-2}$ over those of the synthesized catalysts. The error bar was calculated from three independent tests. (d) Durability test of SnO₂/C (left) and FTO/C (right) catalysts in the flow-type CO₂ electrolyzer in 1 M KHCO_3 solution.

The faradaic efficiencies of CO, H₂ and formate reported were observed during the durability test. (e) FE_{formate} of advanced Sn-based CO₂RR catalysts (f) Plot of the partial current density of formate (mA cm⁻²) versus the durability (h) for various Sn-based CO₂RR electrocatalysts.

Response: For the stability tests, the whole system was considered, that is, both the anodic and cathodic overpotentials and resistances were taken into account. Stability is often gauged from the difference between the initial and final states. From the reviewer's comments, both the local and bulk pH reach a steady state condition during CO₂RR regardless of the initial state of the electrolyte (KOH or KHCO₃) used. Therefore, both these conditions are advantageous for the stability of the total system. During CO₂RR, the cathode experiences localized alkaline pH conditions abruptly in both 1 M KOH and 1 M KHCO₃ solutions. However, alkalinity and CO₂ purging results to the neutralization of the bulk solution. In 1M KOH, neutralization increases the solution resistance, which in turn, significantly increases the overpotential for OER. The potential versus time graph for the anode under 1 M KOH was added. From this plot, it can be seen that the cathodic potential remained constant possibly due to the high local pH (steady state conditions). On the contrary, the anode potential significantly increased after a few hours because of catholyte neutralization. Therein, the catalytic activity was significantly affected by the pH as proven by the substantial increase in the overpotential observed for OER. Therefore, considering the whole system, electrochemical measurements obtained in 1M KHCO₃ were more stable than those recorded using 1 M KOH. The manuscript has been duly revised to clarify this point.

Main manuscript Page 8, Line 24: In 1 M KOH, the anolyte is neutralized during the electrochemical CO₂RR, leading to a high overpotential for OER (Figure S10). Considering the stability of the whole system, the CO₂ electrolyzer was optimized for neutral media.

Figure S10. The anode and cathode potentials (vs Ag/AgCl reference electrode) without iR-correction at a current density of 100 mA cm⁻² in 1 M KOH solution in the flow-type CO₂ electrolyzer.

Comment 2: I cast doubt on the simulation that the formate conversion process involves proton transfer from water to CO₂ since the local pH is alkaline, as revealed by previous reports (ref: Energy Environ. Sci., 2019, 12, 1442-1453). Moreover, the local pH of 1 M KHCO₃ under working conditions is higher than that of the 1 M KOH condition, which was revealed by using in situ observation (J. Am. Chem. Soc. 2020, 142, 15438–15444). Accordingly, the assumption of proton transfer from water to CO₂ is wrong.

Response: Thank you for bringing this to our attention. To address this comment, the elementary steps of the CO₂-to-HCOOH conversion reaction have been revised and the corresponding DFT calculations

were performed. Previous reports tell that, for CO₂ reduction current densities higher than 35 mA cm⁻², the proton is supplied to the adsorbed CO₂ molecules on the electrode surface by the dissociation of water molecules. At this density, a huge amount of unused hydroxide ions is rapidly generated as a by-product, which results to the increase of the local interfacial pH to values above 12. This is regardless of the type of the buffering agent used. In this regard, it can be assumed that the proton for the CO₂ reduction reaction was predominantly supplied by the local electrolyte. The elementary steps of the electrochemical conversion reaction of CO₂ to HCOOH involving 2-electron pathway are as follows:

The free energy diagrams for the CO₂-to-HCOOH conversion on the SnO₂ and FTO (110) surfaces were added in Figure 3. The additional data and revised text for Figure 3 can be found on the “Theoretical investigation of CO₂-to-HCOOH conversion on the SnO₂ and FTO on the ‘surface’” section on page 9 (main text) and on Tables S2 and S3 on page 17 (Supplementary Information). The revised figure and text for the DFT studies are as follows:

Figure 3. Free energy diagram of CO₂ to HCOOH conversion reaction on the (a) SnO₂ (110) and (b) FTO (110) surface (yellow: Sn, red: O, white: H, green: F). The strained supercells were used to simulate the structural change on SnO₂ and FTO nanoparticles at an applied potential of 1.0 V under CO₂ reduction reaction.

For the SnO₂ and FTO (110) surfaces, the CO₂ molecule was gently adsorbed on the Sn atoms. Then, the HCOO* intermediate was formed as the oxygen atoms of CO₂ were tightly bound to the Sn atoms. The difference between the CO₂ adsorption energies on the SnO₂ and FTO (110) surfaces was not substantial. However, approximately, a 1-eV difference was observed on the free energies for the HCOO* intermediate and HCOOH_(g) formation steps. The FTO surface could provide thermodynamically favorable conditions for HCOO* formation compared to the SnO₂ surface, whereas the conditions on the SnO₂ surface favors the HCOOH_(g) formation (Figure 3 and Table S2). From the computational hydrogen electrode model, a minimum potential of -2.05 V was required for SnO₂ to overcome the activation barrier of HCOO* formation, whereas -1.79 V was needed for FTO to complete HCOOH_(g) formation. Identical free energy calculations using strained SnO₂ and FTO supercells were performed to simulate the structural changes on the SnO₂ and FTO electrodes during CO₂ reduction. The in-situ EXAFS data demonstrated that the bonding distance between Sn and Sn (or O) decreased marginally by applying a potential of -1.0 V. Consequently, the lattice constants of the SnO₂ and FTO supercells decreased uniformly by 1.37% and 0.33%, respectively. The observed changes in the lattice

constants correspond to the average change of each bonding distance at -1.0 V. Although the strained SnO_2 (110) surface registered a lower energy for the formation of $\text{HCOOH}_{(g)}$ by 0.42 eV, the energies for CO_2 adsorption and HCOO^* formation were almost identical to those recorded for the SnO_2 surface. As such, the free energy for the HCOO^* formation, which is the potential-limiting step, was not affected by the compression of the SnO_2 (110) crystal. In consequence, a potential of more than 2 V was still required for the strained SnO_2 . In contrast, the strained FTO (110) surface showed a 0.27-eV lower free energy for the $\text{HCOOH}_{(g)}$ formation, while the energies for CO_2 adsorption and HCOO^* formation slightly increased compared to that on the FTO surface. Since the $\text{HCOOH}_{(g)}$ formation is the potential-limiting step for the FTO electrode, the minimum potential to complete the CO_2 -to- HCOOH conversion reaction decreased from -1.79 V to -1.65 V as the FTO crystal was compressed. The DFT studies suggest that doping fluorine to SnO_2 could alter the chemical environment of the oxide surface, making it thermionically stable for HCOO^* intermediate formation, which is an important step in the CO_2 -to- HCOOH conversion reaction. In addition, the compression of the crystal through the application of an external bias rendered an effective contribution only for FTO. We assume that these factors could facilitate formate production in the FTO electrode.

Figure 3. Free energy diagram of CO_2 to HCOOH conversion reaction on the (a) SnO_2 (110) and (b) FTO (110) surface (yellow: Sn, red: O, white: H, green: F). The strained supercells were used to simulate the structural change on SnO_2 and FTO nanoparticles at an applied potential of 1.0 V under CO_2 reduction reaction.

Main manuscript Page 9, Line 30: The enhanced $\text{FE}_{\text{formate}}$ of FTO was investigated through density functional theory (DFT) calculations. As the XRD patterns of the as-prepared SnO_2 and FTO nanoparticles did not reveal any preferred orientations, the SnO_2 and FTO surfaces were modeled using a four-layer slab composed of the tetragonal (110) plane. To build the FTO (110) supercell, 15% of the oxygen atoms in the SnO_2 (110) supercell were randomly replaced with fluorine atoms. The elementary steps of the electrochemical conversion reaction of CO_2 to HCOOH involving two electron pathways were described in three steps (Method). First, CO_2 adsorption on the catalyst surface was performed to consider the onset potential difference between water and CO_2 reduction. The adsorbed CO_2 (CO_2^*) is then converted to the HCOO^* intermediate and $\text{HCOOH}_{(g)}$ in sequence with two proton-coupled electron transfers. A recent study demonstrated that, in CO_2 reduction current densities higher than 35 mA cm^{-2} , the proton can be supplied to CO_2^* on the electrode surface by the dissociation of water molecules. At these current density regions, a huge amount of unused hydroxide ions is rapidly generated as a by-product, which results to the increase of the local interfacial pH to values above 12. This is regardless of the type of the buffering agent used.⁵¹ In this regard, the proton for the CO_2 reduction reaction is assumed to be predominantly supplied by the local electrolyte.

For both the SnO₂ and FTO (110) surfaces, the CO₂ molecule was gently adsorbed on the Sn atoms. Then, the HCOO* intermediate was formed as the oxygen atoms of CO₂ were tightly bound to the Sn atoms. The difference between the CO₂ adsorption energies on the SnO₂ and FTO (110) surfaces was not substantial. However, approximately, a 1-eV difference was observed on the free energies for the HCOO* intermediate and HCOOH_(g) formation steps. The FTO surface could provide thermodynamically favorable conditions for HCOO* formation compared to the SnO₂ surface, whereas the conditions on the SnO₂ surface favors the HCOOH_(g) formation (Figure 3 and Table S2). From the computational hydrogen electrode model, a minimum potential of -2.05 V was required for SnO₂ to overcome the activation barrier of HCOO* formation, whereas -1.79 V was needed for FTO to complete HCOOH_(g) formation. Identical free energy calculations using strained SnO₂ and FTO supercells were performed to simulate the structural changes on the SnO₂ and FTO electrodes during CO₂ reduction. The in-situ EXAFS data demonstrated that the bonding distance between Sn and Sn (or O) was slightly decreased by applying a potential of -1.0 V. Consequently, the lattice constants of the SnO₂ and FTO supercells decreased uniformly by 1.37% and 0.33%, respectively. The observed changes in the lattice constants correspond to the average change of each bonding distance at -1.0 V (Figure S15 and Table S3). Although the strained SnO₂ (110) surface registered a lower energy for the formation of HCOOH_(g) by 0.42 eV, the energies for CO₂ adsorption and HCOO* formation were almost identical to those recorded for the SnO₂ surface. As such, the free energy for the HCOO* formation, which is the potential-limiting step, was not affected by the compression of the SnO₂ (110) crystal. In consequence, a potential of more than 2 V was still required for the strained SnO₂. In contrast, the strained FTO (110) surface showed a 0.27-eV lower free energy for the HCOOH_(g) formation, while the energies for CO₂ adsorption and HCOO* formation slightly increased compared to that on the FTO surface. Since the HCOOH_(g) formation is the potential-limiting step for the FTO electrode, the minimum potential to complete the CO₂-to-HCOOH conversion reaction decreased from -1.79 V to -1.65 V as the FTO crystal was compressed. The DFT studies suggest that doping fluorine to SnO₂ could alter the chemical environment of the oxide surface, making it thermionically stable for HCOO* intermediate formation, which is an important step in the CO₂-to-HCOOH conversion reaction. In addition, the compression of the crystal through the application of an external bias rendered an effective contribution only for FTO. Therefore, these factors are assumed to facilitate formate production in the FTO electrode.

Main manuscript Page 17, Line 29: DFT calculations were performed using the Quantum ESPRESSO package.^{53, 54} Geometry optimization was performed using the Perdew–Burke–Ernzerhof functional⁵⁵ with the projector-augmented wave pseudopotentials.^{56, 57} The Grimme’s D3 (DFT-D3)⁵⁸ method was used to account for the van der Waals dispersion correction. A kinetic energy cutoff of 500 eV was used with a plane-wave basis set. Gaussian smearing was applied with a smearing width of 0.1 eV. The geometries were fully relaxed until the residual force on the atoms converged to 0.01 eV/Å. The SnO₂ (110) supercell was modeled using a unit cell of the tetragonal space group (mp-856) obtained from the Materials Project.⁵⁹ The four layers of the primitive unit cell that are cleaved to the (110) plane were expanded 6 times (2 × 3 × 4), yielding a supercell of 24 Sn and 48 O atoms with a lattice constant of 6.83 Å × 9.73 Å × 13.67 Å. To build the FTO (110) supercell, seven O atoms (~15%) were randomly replaced with F atoms. To simulate the SnO₂ and FTO crystal conditions under an applied potential of -1.0 V, the lattice constants of the SnO₂ and FTO (110) bulk were uniformly reduced by -1.37% and -0.33%, respectively (Figure S15 and Table S1). The lattice constants of strained SnO₂ and FTO (110) bulk supercell were 6.74 Å × 9.60 Å × 13.48 Å and 6.81 Å × 9.70 Å × 13.62 Å, respectively. The Brillouin zone was sampled with a Monkhorst–Pack k-point mesh of (5 × 4 × 2) for bulk SnO₂ and FTO (110) supercells. To build the slab structure, a 20-Å vacuum gap was added along the c-axis of the stabilized supercells. The bottom two layers of the slab were fixed, while the top two layers were allowed to relax during geometry optimization. The Brillouin zone was sampled with a Monkhorst–Pack k-point mesh of (5 × 3 × 1) for SnO₂ and FTO (110) slab supercells.

The elementary steps of the electrochemical conversion reaction of CO₂ to HCOOH involving 2 electron pathway are described as follows:

where * represents the surface sites for molecule adsorption. The change in Gibbs free energy (ΔG) at 298 K and 1 atm was calculated thorough $\Delta G = \Delta E + \Delta ZPE - T\Delta S$, where ΔE is the total electronic energy obtained from the DFT optimization, ΔZPE is the change in the zero-point energies, T is the temperature, and ΔS is the change in entropy. The computational hydrogen electrode model⁶⁰ was applied to calculate the chemical potential of proton/electron pairs, which is equal to the half of the chemical potential of H₂ gas under standard conditions and electrons with an applied bias of U ($-eU$). The pH contribution is considered by adding $k_B T \times \ln 10 \times \text{pH}$ to ΔG , where k_B is the Boltzmann constant. The ZPE and S of the molecules and adsorbates were determined from the calculated vibrational frequencies and NIST database,^{61, 62} where all vibrations were treated in the harmonic oscillator approximation. The ZPE and S data are listed in Tables S2 and S5, respectively.

Table S2. DFT calculated adsorption energy (ΔE), zero-point energy (ZPE), and entropy (S) at 298 K and 1 atm of CO₂ and HCOO adsorbate on the slab surface.

	Adsorbed surface	ΔE (eV)	ZPE (eV)	TS (eV)
CO ₂ *	SnO ₂ (110)	-0.347	0.312	0.020
	SnO ₂ (110), -1.37% strained	-0.354	0.320	0.025
	FTO(110)	-0.091	0.309	0.037
	FTO(110), -0.33% strained	0.112	0.308	0.033
HCOO*	SnO ₂ (110)	-2.649	0.686	0.169
	SnO ₂ (110), -1.37% strained	-2.692	0.689	0.174
	FTO(110)	-3.484	0.666	0.193
	FTO(110), -0.33% strained	-3.235	0.661	0.189

Table S5. Zero-point energy correction and entropy contribution of H₂, CO₂, and HCOOH gas at 298 K and 1 atm.

	ZPE (eV)	TS (eV)
H ₂	0.276	0.441
CO ₂	0.308	0.656
HCOOH	0.854	1.071

Comment 3: For the theoretical calculations, the results should be based on the experimental evidence obtained from the in-situ observations since the catalyst undergoes structural evolution during the reaction, as revealed by the in situ XAS and Raman results. As a result, I will suggest the authors should provide a correct model based on the in-situ measurements.

Response: Thank you for your insightful comments. To address this point, the average bonding distances of Sn-Sn and Sn-O in the SnO₂ and FTO crystals at an applied potential of -1.0 V were obtained

from the in-situ EXAFS data. Then, DFT calculations using the strained SnO₂ and FTO supercells were performed. The lattice constants of SnO₂ and FTO supercells decreased uniformly by -1.37% and -0.33% based on the EXAFS fitting data. The free energy diagrams of CO₂-to-HCOOH conversion on the strained supercells were added to the revised Figure 3. The thermodynamic activation barrier for CO₂-to-HCOOH conversion between strained SnO₂ and the SnO₂ (110) surface was almost identical. On the other hand, a large decrease in the activation barrier to complete the HCOOH conversion reaction was observed for the strained FTO (110). The free energy diagram and detailed analysis are described in Section 2.

Comment 4: The authors should provide EXAFS results extracted from XAS. I believe that if F could stabilize Sn, the local structure may retain the initial situation.

Response: We appreciate the reviewer's helpful comments. EXAFS analyses for the SnO₂ and FTO catalysts were conducted. The corresponding results are shown in Figure S15 and summarized in Table S3. As mentioned in the response to comment 3, the change in the lattice parameter of FTO is significantly less than that observed on SnO₂.

Figure S15. Fourier transforms of k^3 -weighted Sn L_{III}-edge EXAFS for (a) SnO₂ and FTO catalyst obtained by in-situ/operando hard-XAS analysis

Table S3. EXAFS fitting results of SnO₂ and FTO catalysts. The multi-shell fitting results ($\Delta k = 2 \sim 12 \text{ \AA}^{-1}$) of the experimental EXAFS spectrum

Catalyst	Condition	Sn-Sn bond	Sn-F bond	Sn-O bond
SnO ₂	ex-situ	3.2595	-	2.0762
	-1.0 V	3.2063	-	2.0541
FTO	ex-situ	-	2.0359	2.0975
	-1.0 V	-	2.0305	2.0905

Main manuscript Page 10, Line 27: The in-situ EXAFS data demonstrated that the bonding distance between Sn and Sn (or O) was slightly decreased by applying a potential of -1.0 V. Consequently, the lattice constants of the SnO₂ and FTO supercells decreased uniformly by 1.37% and 0.33%, respectively. The observed changes in the lattice constants correspond to the average change of each bonding distance at -1.0 V (Figure S15 and Table S3).

Main manuscript Page 17, Line 10: For the EXAFS analysis, Artemis (also implemented in Demeter ver. 0.9.20) software was utilized after the processing of data using the Athena software. The background signal was removed to extract the EXAFS signal for $R_{\text{bkg}} = 1-1.1 \text{ \AA}$. The EXAFS data were transformed using the Kaiser-Bessel function. The many-body reduction factor (S_0^2) for Sn was determined to be 0.86 from the EXAFS curve fit of the Sn foil. The statistical quality of the curve fit to the proposed models can be determined from the R-factor and χ^2 function available in the refinement.

Comment 5: The authors suggested that the valence state of Sn of the FTO catalyst is oxidized under high cathodic potentials. However, according to previous reports, the oxidized feature under cathodic potential may result from the sluggish structural transformation of Sn. It may transform into metallic Sn under long-term observation. Based on this reason, I think the authors should provide a long-term observation to ascertain the oxidized Sn is stable under cathodic conditions.

Response: Thank you for this valuable comment. The stability of the oxidation state of the FTO catalyst is a key consideration in this study. As such, XPS results and EDS-TEM images after the stability test were added. The oxidation states and structure of the FTO catalyst are well maintained after the stability tests under long-term cathodic conditions. These results suggest that F increase the stability of FTO catalyst under high cathodic potentials.

Main manuscript Page 8, Line 32: X-ray photoelectron spectroscopy (XPS), HAADF-STEM, and EDS after the stability tests demonstrated that the structure and chemical state of FTO/C remained the same even after the exposure to long-term cathodic conditions (Figure S13 and S14).

Figure S13. XPS spectra of Sn 3d on FTO/C (a) before and (b) after stability tests

Comment 6: In my opinion, the FTO/C catalyst would transform into metallic Sn after days of operation. Based on this assumption, only a trace amount of fluorine could be detected after the stability test. To resolve my concern, structural characterization such as HAADF-STEM and EDS results of post-reaction catalysts should be provided. If fluorine of FTO/C could be detected by using EDS after the reaction, it could help the authors to strengthen their claim.

Response: Thank you for your insightful comments. Following the reviewer's advice, EDS mapping images of the FTO catalyst after the stability test were added accordingly. Fluorine was detected even after 7 days of operation using the FTO catalyst.

Main manuscript Page 8, Line 32: X-ray photoelectron spectroscopy (XPS), HAADF-STEM, and EDS analyses after the stability tests demonstrated that the structure and chemical state of FTO/C remained the same even after the exposure to long-term cathodic conditions (Figure S13 and S14).

Figure S14. (a) HAADF-STEM image and its energy dispersive X-ray spectroscopy (EDS) mapping images of (b) Sn (red), (c) F (magenta) and (d) layered image combining all 3 maps for FTO/C after stability test. The signal collecting time was 5 minutes.

Comment 7: The preparation procedure of SnO₂ and doped-SnO₂ catalysts is ambiguous. The authors should provide the exact formula in the experimental section.

Response: We appreciate the reviewer's helpful comments. First, the formation of the tin ethbutoxide intermediate in the ethanol solution and the formation of SnO₂ through the hydrolysis reaction are described in Statement S1. Antimony and indium were doped in the form of mixing Sb₂O₃ or In₂O₃ with SnO₂ during hydrolysis and condensation, respectively. On the other hand, fluorine was incorporated through oxygen substitution. In order to address the reviewer's advice, a more detailed discussion on the preparation method for the doped-SnO₂ catalysts was added in the Experimental Methods section.

Main manuscript Page 14, Line 11: A detailed description of the formation of the tin ethbutoxide intermediate in the ethanol solution and the formation of SnO₂ through the hydrolysis reaction is described in Statement S1.

Main manuscript Page 14, Line 16: Varying amounts of the dopant salts were added to 712 mg of SnCl₄ to synthesize the doped-SnO₂ sample. In ATO, 25.0 mg C₆H₅O₆Sb was used, in FTO, 19.48 g NH₄F, and in ITO, 109.6 g InCl₃. Antimony and indium were doped in the form of mixing Sb₂O₃ or In₂O₃ with SnO₂ during hydrolysis and condensation, respectively. On the other hand, fluorine was incorporated through oxygen substitution.

Comment 8: According to LSV and FE results, SnO₂ with TDA during the synthesis process shows higher activity and selectivity toward formate. The authors claim that the particle size and uniformity of SnO₂ have significant effects on the CO₂RR activity. However, the particle size of SnO₂ with/without TDA is similar. I consider that the authors should provide more evidence such as TEM images for post-reaction catalysts to ascertain the claim for the contradiction.

Response: Thank you for your insightful comments that help us improve the manuscript. Following the reviewer's advice, TEM images of the SnO₂ catalysts with and without TDA surfactant after the CO₂RR were added. From these images, there was no observed substantial change in the particle size before and after CO₂RR. Without TDA, the SnO₂ particles are agglomerated. The difference in CO₂RR activity between the SnO₂ catalysts synthesized with and without TDA can be attributed to the difference in uniformity, which is more pronounced than the difference in particle size. The related paragraph has been modified as follows:

Main manuscript Page 8, Line 7: The effect of TDA on the SnO₂ particle size before and after CO₂RR is not substantial. However, without TDA, the SnO₂ particles are agglomerated (Figure S9: HR-TEM images of the SnO₂ catalyst with and without TDA). These results show that the uniformity of the SnO₂ particles significantly affects their CO₂RR activity.

Figure S9. HR-TEM images of SnO₂ (a) with and (b) without TDA surfactant after CO₂RR

To sum up, I think that the author should prove Sn is in a high oxidation state and fluorine retain under long-term examination. Moreover, the DFT calculation should be revised since it is based on an inappropriate model. As for present manuscript, this research falls short of being published in Nat. Commun.

Reviewer: 3

Recommendation: This manuscript deals with the preparation and characterization of SnO₂ catalysts doped with fluorine, which are then assessed when used in electrodes for the electrochemical reduction of CO₂ to formate. The most relevant results are related to the improved performance of fluorine-doped SnO₂ electrocatalysts, which show excellent Faradaic efficiencies at reasonably high current density (100 mA cm⁻²) during 7 days of operation. The manuscript addresses a hot topic, and it may report interesting new results, including observations from in situ/operando spectroscopies. However, I feel there are some aspects and mistakes that deserve clarification and improvement before it could be eventually accepted for publication:

Comment 1: Page 3, lines 75-76: “Various studies have been devoted to the electrocatalytic reduction of CO₂ to formate,⁷⁻¹¹”. Why have references 7-11 been selected as “studies devoted to the electrocatalytic reduction of CO₂ to formate?” Some brief justification of their relevance should be provided to show they have not been chosen randomly.

Response: We are thankful to the reviewer for this valuable comment. The studies we referred to are on CO₂RR to formate using tin-based electrocatalysts. In particular, Ref. 7, 8, 10, and 11 reported the use of heteroatom-doped or alloy catalysts, whereas Ref. 9 used catalysts that have undergone post-treatment to increase their active sites. We have added the following relevant sentences to the manuscript.

Main manuscript Page 3, Line 13: Various studies on the use of heteroatom-doped/alloy catalysts or catalyst structures with large active areas for the electrocatalytic reduction of CO₂ to formate have been reported.^{7, 8, 9, 10, 11} However, a highly efficient catalyst that can meet commercial requirements for activity, selectivity, and stability has not yet been achieved.

Comment 2: Page 3, line 76: “...highly efficient catalyst...”: Great emphasis is placed here on the nature of the catalyst, which has indeed a great influence. However, some recent studies also show that not just the nature of the catalysts, but also the configuration of the electrode (*i.e.* the form in which it is used in the cathode) and the reactor design can play an important role in the performance of electrocatalytic reduction of CO₂ (see e.g. Chem. Eng. J. 364 (2019) 89–101; Chem. Eng. J. 405 (2021) 126965). It could be interesting to strengthen the introduction briefly discussing this idea.

Response: We are thankful to the reviewer for this valuable comment. According to the comment, device design significantly affects the performance of the catalysts towards the electrocatalytic reduction of CO₂. In this work, a gas-type flow cell as used to achieve a high current density. The importance of device design has been discussed in the introduction.

Main manuscript Page 4, Line 15: The electrochemical CO₂RR performance is significantly affected by the device design and the type of purged CO₂.^{44,45} Therefore, a homemade gaseous CO₂ fed flow cell was used to achieve a high current density. *In-situ/operando* analysis was conducted under similar conditions.

Comment 3: Page 3, lines 89-90: “Sn is most reasonable materials because the toxicity (Pb, Hg) or high price (Bi) of catalyst materials limits commercialization”. Have the authors checked that the price of Bi particles is really higher than Sn particles? A quick check in suppliers like Merck or Sigma-Aldrich shows that the price of Bi powders per gram is similar, or even lower, than the price of Sn powder of the same particle size, and if I am not wrong, the cost of Sn per kg is several times higher than that of Bi. Authors should carefully check this, and if necessary, correct the statement.

Response: Thank you for bringing this to our attention. In the past year, the price of Sn has more than doubled, making it cost as much as Bi (Reference:

<https://www.dailymetalprice.com/metalpricecharts.php?c=sn&u=lb&d=480>). Therefore, the phrase was modified to emphasize the larger deposits of Sn than Bi in the Earth's crust, rather than its lower cost.

Main manuscript Page 3, Line 30: Among these materials, Sn is the most reasonable material because the toxicity (Pb, Hg) or relatively scarcity in the earth's crust (Bi) of other typical catalyst materials limit their commercialization.²⁶

Comment 4: Page 14, lines 363-364: In the "Preparation of SnO₂-based catalyst electrodes for the CO₂RR", it is only stated that "A catalyst ink was prepared by ultrasonically mixing 5 wt% of ionomer solution (Dioxide, 15 wt% target of catalyst)..." but the nature of the ionomer was not clear to me. Which compound was used as binder and why?

Response: We are thankful to the reviewer for this valuable comment. In flow cells, the binders are used to ensure the adhesion of the catalysts on the electrode. They do not critically act to transport the ions. It is important that the binder holds the catalyst well without blocking the ions for CO₂RR. In this study, the Dioxide AEM and ionomer, which are produced from the dioxide, were used. This dioxide ionomer has a similar property with the dioxide AEM.

Comment 5: Page 14, lines 366-367: Also regarding the preparation of the SnO₂-based cathodes, the loading of catalyst used can play an important role in the performance of the electrode. How and why was the loading fixed at 0.5 mg cm⁻²? This is an important aspect that deserves further explanation.

Response: Thank you for this comment. Catalyst loading is an important factor for the electrocatalytic performance of the electrode. The performance of the SnO₂-based electrode decreased as the catalyst loading decreased below 0.5 mg cm⁻². On the other hand, similar CO₂RR performances were recorded at 0.5 and 1 mg cm⁻². Therefore, the catalyst loading was fixed at 0.5 mg cm⁻² because of the good CO₂RR performances registered using a favorable amount of catalyst.

Comment 6: Different electrodes were used as anodes for the OER in neutral and in alkaline medium. It could be interesting to include, in the Supporting Information, the reasons to justify this decision.

Response: We appreciate the reviewer's valuable comments. Fe-Ni foam exhibits excellent OER performance under alkaline conditions. At the same time, it is cheaper than an iridium catalyst. However, it slowly dissolves in neutral conditions, leading to the degradation of the OER performance of the electrode. Furthermore, the dissolved Ni ion can cross over to the cathode, reducing the FE for CO₂RR. Therefore, IrO₂/Pt coated Ti-foam electrodes were used for the OER in neutral media. This explanation has been added to the Supporting Information.

Statement S2. The reason for the use of different OER electrodes in neutral and alkaline electrolytes.

Fe-Ni foam exhibits excellent OER performance under alkaline conditions. At the same time, it is cheaper than an iridium catalyst. However, it slowly dissolves in neutral conditions, leading to the degradation of the OER performance of the electrode. Furthermore, the dissolved Ni ion can cross over to the cathode, reducing the FE for CO₂RR and inducing the hydrogen evolution reaction (HER). Therefore, IrO₂/Pt coated Ti-foam electrodes were used for the OER in neutral media for a stable CO₂RR operation.

Comment 7: It may difficult for the reader to know exactly the different catalysts and electrodes that were tested in the study. I suggest to clearly identify them in "4. Experiment methods", at the beginning of subsection "Electrochemical CO₂RR flow cell tests".

Response: Thank you for bringing this to our attention. A table summarizing the catalysts used in the device experiments was added in the Supporting Information.

Table S4. Summary of electrocatalysts, electrolyte and formate electrocatalytic activity for all device tests.

Cathode	Anode	Electrolyte	Maximum Formate F.E (%)	Maximum Formate partial current density (mA cm ⁻²)
SnO ₂ /C w/o TDA	Fe-Ni foam	1 M KOH	76	76
SnO ₂ /C with TDA	Fe-Ni foam	1 M KOH	80	215
	IrO ₂ /Pt coated Ti-foam	1 M KHCO ₃	75	161
FTO/C with TDA	Fe-Ni foam	1 M KOH	95	330
	IrO ₂ /Pt coated Ti-foam	1 M KHCO ₃	89	222
ATO/C with TDA	Fe-Ni foam	1 M KOH	80	242
ITO/C with TDA	Fe-Ni foam	1 M KOH	85	272

Comment 8: Page 14, lines 387-388: The schematic of the flow cell is not in Figure S6, but in Figure S7. Please correct.

Response: Thank you for pointing this out. The text has been revised as follows to reflect this correction.

Main manuscript Page 15, Line 15: A detailed schematic of the flow cell used for evaluating the electrochemical CO₂RR performance is shown in Figure S7.

Comment 9: Page 14, lines 388-389: “The fabricated SnO₂-based catalyst electrodes were used as both the cathode and anode.” What do the authors mean with “both as cathode and anode”? How were the Sn-based electrodes used as anodes?

Response: Thank you for pointing this out. The sentence was corrected in the revised manuscript, according to your advice.

Main manuscript Page 15, Line 15: The fabricated SnO₂-based catalyst electrodes were used as the cathode.

Comment 10: The way formate was collected and analyzed was not clear to me. In page 15 it is only mentioned that “IC (...) was used to monitor formate (...)”. But which liquid streams did the authors actually analyze? Was formate obtained in the anolyte stream, in the catholyte or in both? My concern is that the use of an anion exchange allows the pass of anions, which means that HCOO⁻ formed in the cathodic compartment should cross the membrane and be collected in the anolyte, but perhaps only partially, and some formate product may also be found in the catholyte. I feel this should be clearly clarified in the manuscript.

Response: Thank you for your valuable comment. As per the reviewer's comment, formate can cross over to the anodic side. Thus, samples for IC at both liquid streams were collected both from the anode and the cathode side. In this work, small amounts of formate were detected in the anolyte stream. We have revised the manuscript to address this point.

Main manuscript Page 15, Line 24: The formate concentrations on the catholyte and anolyte were measured to calculate the total production of formate.

Comment 11: Page 16, line 433. "...is shown in Figure XX." Please revise.

Response: Thank you for pointing this out. We have added a figure for the *in-situ/operando* hard-XAS measurement setup and corrected the relevant sentences in the revised manuscript.

Main manuscript Page 16, Line 32: The setup for the *in-situ/operando* hard-XAS measurements with a homemade electrochemical single cell is shown in Figure 16.

Figure S16. System of *in-situ/operando* hard-XAS analysis with the homemade electrochemical zero-gap device.

Comment 12: Page 17. The selection of the most appropriate functionals and potentials for the DFT calculations is not trivial. How and why were these chosen for this specific application?

Response: We are thankful to the reviewer for this valuable comment. We strongly agree with your comment that the appropriate selection of functionals and potentials is crucial for obtaining correct and accurate results from the DFT calculations. We have performed all DFT calculations to address Reviewer 1's suggestion with the DFT-D3 functional and PAW pseudopotentials (PP). This combination is widely used in studies on heavy solid surfaces describing adsorptions of light molecules, as well as HER (Nat Commun 2020, 11, 3315), OER (Nature Commun 2018, 9, 2885), and CO₂RR (Nat Commun 2019, 10, 1875). Although the dispersion correction with the DFT-D3 functional using pre-calculated parameters can be less accurate than van der Waals density functionals such as vdW-DF, this functional has shown fairly accurate results in both molecular and solid-state applications (Phys Chem Chem Phys 2015, 17, 6423; Chem Rev 2016, 116, 5105). In addition, the DFT-D3 method did not add a significant computational cost, whereas the vdW-DF approach increased the computational time by approximately 50% compared to the GGA calculation (J Chem Phys 2012, 137, 120901). This enabled us to investigate a diverse combination of molecular adsorption at a variety of surface planes. The PAW PP is also widely used for the same reasons as DFT-D3. The norm-conserving (NC) PP could provide more accurate results than PAW PP. NC PP describes the actual radial orbitals of atoms better than PAW PP. However,

a high kinetic energy cut-off of more than 1000 eV is required to obtain accurate data with the NC PP. For practical reasons, we performed DFT calculations with DFT-D3 functional and PAW PP, and we ensured that our selection is appropriate because the free energy diagrams of CO₂-to-HCOOH conversion (Figure 3) clearly represent the catalytic activity difference between the SnO₂ and FTO surfaces.

Comment 13: Figure 1b and c: Considering that, according to the insets with particle size distributions, the average particle size seems to be around 2 nm, are scales of 10 or 20 nm the most adequate to show?

Response: Thank you for bringing this to our attention. Scale bars (5 nm) were added to Figures 1b and c in the revised manuscript.

Figure 1. Physical properties and synthetic scheme of Sn-based catalysts. (a) Synthetic pathway of well-dispersed Sn-based catalysts supported on carbon: (1) formation of metal-surfactant complex, (2) hydrolysis and condensation, (3) formation of micelle-like surfactant templates, and (4) hydrothermal treatment for recrystallization. HR-TEM images of (b) SnO₂/C and (c) FTO/C catalysts (Inset: particle size distributions; average particle sizes and standard deviations fitted with a Gaussian function). (d) HAADF-STEM image and its energy dispersive X-ray spectroscopy (EDS) mapping images of Sn (red), F (magenta) and layered image combining all maps for FTO/C. The signal collecting time was 5 minutes. (e) Powder XRD spectra of Sn-based catalysts with various dopants. It indicated that the no phase change occurred by dopants.

Comment 14: Figure 2 and Figure S7. As shown in the schematic illustration, in the anodic compartment, an “Oxygen Evolution Electrode” and a “Titanium Plate” are represented. Why is the titanium plate used? If there is already an OER electrode, which, as described in page 14, already includes metallic foams that can act as current collector, why is the additional titanium plate needed?

Response: We appreciate the reviewer's valuable comments. The metal foam electrode and the titanium plate were used as the catalyst electrode and current collector, respectively. An electrode can be used as a current collector. However, an additional flow plate and contact line were necessary in this case, hence the use of an additional titanium plate.

Comment 15: Fig S8: Particularly at the highest current density studied (400 mA cm^{-2}), the sum of FEs is further from approaching 100% than at lower current densities. Do the authors have any explanation for that?

Response: Thank you for this valuable comment. This is an important issue. In the HER dominant region, i.e., at the high current density region, the reproducibility is decreased, and so is the sum of FE. Since the catalyst is made for CO_2RR , HER is expected to be non-uniform. Furthermore, the line gas flow rate, which also affects the total FE, is also non-uniform in the HER region. However, since the problem does not considerably affect the formate production rate, it does not have implications on the insights of this study. We are aware of this issue and we will look at this further in our future experiments.

Comment 16: Another important concern is related to the reproducibility of the results obtained. For example, were the experiments in the flow cell reported in Figure 2 carried out more than once? In that case, which values of deviation were observed *e.g.* in the results of Faraday efficiencies and production rates? Some indication of the reproducibility of the results (*e.g.* including error bars with standard deviation) must be provided.

Response: We are thankful to the reviewer for this valuable comment. Reproducibility is an important consideration in scientific works. As such, the reproducibility of the results reported herein was assessed by conducting the measurements thrice for each sample. An error bar for the FE values obtained for formate was added in Figure 2c. The results showed good reproducibility below 300 mA cm^{-2} . However, the error increased at high current density measurements at the HER dominant region.

Figure 2. Single-cell performances of Sn-based catalysts. (a) Schematic of the flow-type CO₂ electrolyzer using a gas-diffusion layer. (b) Faradaic efficiencies of the products and production rates of formate for FTO/C catalyst at each given current density in 1 M KOH solution. (c) Partial current densities of formate for CO₂RR in the current density range of 50-400 mA cm⁻² over those of the synthesized catalysts. The error bar was calculated from three independent tests. (d) Durability test of SnO₂/C (left) and FTO/C (right) catalysts in the flow-type CO₂ electrolyzer in 1 M KHCO₃ solution. The faradaic efficiencies of CO, H₂ and formate reported were observed during the durability test. (e) FE_{formate} of advanced Sn-based CO₂RR catalysts (f) Plot of the partial current density of formate (mA cm⁻²) versus the durability (h) for various Sn-based CO₂RR electrocatalysts.

Comment 17: The way the experimental results are presented and discussed on “CO₂-to-formate conversion performance” should be improved. In line with a previous comment, different types of catalysts, and even different conditions are mentioned (*e.g.* type of electrolyte: KOH or KHCO₃), but I feel that the global picture of which combinations were actually tested was not clearly shown. Perhaps, a good way to clearly show this, could be to include in the Supporting information a Table with all the tests carried out with their different experimental conditions and results.

Response: Thank you for bringing this to our attention. Following the reviewer’s advice, we added a table summarizing the details of the CO₂RR experimental conditions in the revised Supporting Information.

Table S4. Summary of electrocatalysts, electrolyte and formate electrocatalytic activity for all device experiment.

Cathode	Anode	Electrolyte	Maximum Formate F.E (%)	Maximum Formate partial current density (mA cm ⁻²)
SnO ₂ /C w/o TDA	Fe-Ni foam	1 M KOH	76	76
SnO ₂ /C with TDA	Fe-Ni foam	1 M KOH	80	215
	IrO ₂ /Pt coated Ti-foam	1 M KHCO ₃	75	161
FTO/C with TDA	Fe-Ni foam	1 M KOH	95	330
	IrO ₂ /Pt coated Ti-foam	1 M KHCO ₃	89	222
ATO/C with TDA	Fe-Ni foam	1 M KOH	80	242
ITO/C with TDA	Fe-Ni foam	1 M KOH	85	272

Comment 18: The “formate production rate” (page 194, figure 2 and others) is expressed as “mg h⁻¹ cm⁻²” But does the mass unit represent only formate anion (HCOO⁻) or the corresponding salt? To avoid any ambiguity, I suggest to convert all these values and report the production rates with units of “mol”.

Response: We are thankful to the reviewer for this valuable comment. The mass unit (mg h⁻¹ cm⁻²) represents only that of formate anion (HCOO⁻). Accordingly, the unit for formate production rate was duly revised from mg h⁻¹ cm⁻² to mmol h⁻¹ cm⁻² in the manuscript.

Figure 2. Single-cell performances of Sn-based catalysts. (a) Schematic of the flow-type CO₂ electrolyzer using a gas-diffusion layer. (b) Faradaic efficiencies of the products and production rates of formate for FTO/C catalyst at each given current density in 1 M KOH solution. (c) Partial current densities of formate for CO₂RR in the current density range of 50-400 mA cm⁻² over those of the synthesized catalysts. The error bar was calculated from three independent tests. (d) Durability test of SnO₂/C (left) and FTO/C (right) catalysts in the flow-type CO₂ electrolyzer in 1 M KHCO₃ solution. The faradaic efficiencies of CO, H₂ and formate reported were observed during the durability test. (e) FE_{formate} of advanced Sn-based CO₂RR catalysts (f) Plot of the partial current density of formate (mA cm⁻²) versus the durability (h) for various Sn-based CO₂RR electrocatalysts.

Figure S8. (a) LSV curve for SnO₂/C catalysts with and without TDA surfactant and FE of products and production rates of formate for (b) SnO₂/C without TDA surfactant, (c) SnO₂/C with TDA surfactant, (d) ATO/C and (e) ITO/C catalysts at each given current densities. Reaction conditions: 1 M KOH solution (alkaline condition)

Comment 19: Line 219, Table S1, figures 2e and 2f: Which were the criteria for considering particularly these studies using Sn-based electrodes? There are many other relevant studies in the literature using Sn-based electrodes for the electroreduction of CO₂ to formate that could have also been included, so again, the criteria used for selecting these references in Table S1 should be explicitly provided to show they have not been chosen randomly.

Response: We appreciate the reviewer's valuable comments. Many studies have reported the use of Sn-based electrodes for the catalysis of CO₂RR. Thus, the results of the Sn-based catalyst should be selected. Herein, papers that reported Sn-based catalysts with good activity and stability were chosen. The results of such were summarized in Table S1 and depicted in Figure 2e and 2f. In Figure 2e, various representative catalysts with different FE are shown, whereas in Figure 2f, different catalysts with good current density and durability are depicted. There were some results with overlapping performance and durability, thus materials with interesting catalytic properties were chosen.

REVIEWERS' COMMENTS

Reviewer #1 (Remarks to the Author):

he authors revised substantially the MS based on the reviewer's suggestions. Therefore, it can be accepted for publication.

Reviewer #2 (Remarks to the Author):

The authors have thoroughly addressed my concerns.
However, minor issues such as unifying the reference format should be revised.

Reviewer #3 (Remarks to the Author):

The comments of the previous revision have been properly answered, and the manuscript has been improved accordingly. Therefore, in my opinion the paper may be accepted for publication.